# An open-source radar-based hail damage model for buildings and cars

Timo Schmid[1,2], Raphael Portmann[3], Leonie Villiger[1,2], Katharina Schröer[4], and David N. Bresch[1,2]

[1]Institute for Environmental Decisions, ETH Zurich, Zurich, Switzerland
[2]Federal Office of Meteorology and Climatology MeteoSwiss, Zurich, Switzerland
[3]Agroscope Reckenholz, Zurich, Switzerland
[4]Faculty of Environment and Natural Resources, Albert-Ludwigs-University Freiburg, Freiburg, Germany
**Correspondence:** Timo Schmid (timo.schmid@usys.ethz.ch)

**Abstract.** Severe hailstorms cause substantial damages to buildings and vehicles, necessitating the quantification of associated risks. Here, we present a novel open-source hail damage model for buildings and cars based on single-polarization radar data and 250'000 geolocated hail damage reports in Switzerland from 2002 to 2021. To this end, we conduct a detailed evaluation of different radar-based hail intensity measures at 1 km resolution and find that the maximum expected severe hail size (MESHS) outperforms the other measures, despite a considerable false alarm ratio. Asset-specific hail damage impact functions for buildings and cars are calibrated based on MESHS and incorporated into the open-source risk modelling platform CLIMADA. The model successfully estimates the correct order of magnitude for the number of building damages in 91%, their total cost in 77%, the number of vehicle damages in 74%, and their total cost in 60% of over 100 considered large hail events. We found considerable uncertainties in hail damage estimates, which are largely attributable to limitations of radar-based hail detection. Therefore, we explore the usage of crowdsourced hail reports and find substantially improved spatial representation of severe hail for individual events. By highlighting the potential and limitations of radar-based hail size estimates, particularly MESHS, and the utilization of an open-source risk modelling platform, this study represents a significant step towards addressing the gap in risk quantification associated with severe hail events in Switzerland.

# 1 Introduction

Severe hail storms constitute one of the leading damage drivers for buildings and cars in Switzerland, with a recent extreme event on 28 June 2021 (see also Kopp et al., 2022) causing building damages of 400 million Swiss francs (CHF) in a single canton (GVL, 2022), with 1 CHF corresponding to 1.09 US dollars and 0.92 EURO in the reference year 2021 (OECD, 2023). A potential increase in hail severity under climate change (Raupach et al., 2021) may further exacerbate such hail damages in the future. Structural damages to the roof and blinds are typically caused by hailstones with diameters larger than $2.5\,\mathrm{cm}$ (Stucki and Egli, 2007) and dents in the vehicle body from $2\,\mathrm{cm}$ (Hohl et al., 2002b). However, large damages such as destroyed roof tiles or broken car windows are only expected for hailstones greater than approx. 4–5 cm (Yeo et al., 1999; Heymsfield and Wright, 2014; Púčik et al., 2019).

Hail is formed when supercooled water is collected by a hail embryo (e.g. frozen drop) in the updraft of a convective storm (Allen et al., 2020). A key factor for the final hail size is the residence time of a hailstone within the hail growth zone and the supply of supercooled liquid in the thundercloud. Thus, hailstone sizes are primarily limited by the updraft speed in a thunderstorm cell which determines how large hailstones can grow before their fall speed exceeds the upwind velocity. However, this dependence is not a direct relationship because the spatial structure of local updrafts and horizontal winds in each individual thunderstorm influence the trajectory of hailstones (Kumjian and Lombardo, 2020).

The local hail intensity on ground is primarily expressed by the maximum hailstone size or the hail kinetic energy ($\mathrm{E}_{kin}$), although other measures such as the number concentration or the ice mass also exist (Grieser and Hill, 2019). Neither $\mathrm{E}_{kin}$ nor hailstone sizes can be measured directly, except for point measurements with hail pads (Punge and Kunz, 2016), hail sensors (Kopp et al., 2023), or verified crowdsourced reports, e.g. from the European Severe Weather Database (ESWD; Dotzek et al., 2009). Thus, for data with consistent spatial and temporal coverage, we rely on radar measurements. When using radar-based hail intensity estimates, it is important to know that hail stone sizes can approach or exceed radar wavelengths leading to resonance scattering effects (e.g. Bohren and Battan, 1982). Additionally, individual stones can have complex shapes with many side lobes, and may be coated with water (Atlas et al., 1960), which makes a direct hail size estimate from operational weather radars challenging.

Radar-based estimates of hail intensity in Switzerland include the probability of hail (POH) and maximum expected severe hail size (MESHS), which are based on an algorithm originally derived in Waldvogel et al. (1979) depending on the height between the $0\,^{\circ}\mathrm{C}$ level and the highest elevation with a radar reflectivity threshold of 45 and 50 dBZ, respectively. Furthermore, hail kinetic energy has been estimated directly from radar reflectivity (Waldvogel et al., 1978; Hohl et al., 2002a). The radar network of MeteoSwiss (Fig. 1) consisted of three single-polarized C-band radars (Albis, Monte Lema, La Dôle) until it was expanded by two more (Plaine Morte, Weissfluh) in 2012. Also in 2011/12, all five radars were equipped with dual-polarization technology, which lead to improved clutter filtering (Germann et al., 2015) and the development of a hydrometeor classification algorithm that includes hail (Besic et al., 2016, 2018), but no hail intensity metric based on dual-polarization data is available to date.

In other European countries with C-band radar networks, it has been shown that hail can successfully be detected with the criterion of Waldvogel et al. (1979), on which also POH and MESHS are based. Puskeiler et al. (2016) found the highest skill in detecting hail days over 7 years for an adjusted version of the Waldvogel et al. (1979) criterion, compared to a simple reflectivity threshold of 55 dBZ (Mason (1971) criterion). Analyzing one summer in the Netherlands, Holleman et al. (2000) found the Waldvogel et al. (1979) criterion to best estimate the probability of hail compared to Vertically Integrated Liquid (VIL; Amburn and Wolf, 1997), the severe hail index (SHI, see below), the Mason (1971) criterion, and a cloud-top temperature-based method by Auer (1994). Skripniková and Řezáčová (2014) developed a combination criterion of the Waldvogel et al. (1979) method, the SHI (see below), and a SHI-derived quantity called "probability of severe hail" to best detect large hail (>2 cm) in Germany and the Czech Republic from 2002-2011.

In the US, with an operational S-band radar network, the SHI (Witt et al., 1998) and the derived "maximum estimated size of hail" (MESH) are widely used. The SHI is calculated as the vertical integral of a reflectivity-based hail kinetic energy flux estimate (Waldvogel et al., 1978) weighted by the height of the reflectivity in relation to the 0 and -20 °C height.

Accurate hail damage modelling is crucial for insurance companies, resulting in many proprietary models and few openly available studies on the topic. Hohl et al. (2002a, b) investigated relationships of radar-derived hail kinetic energy to car and building damage data per Swiss community level. The authors derived vulnerability curves using the relationship of radar reflectivity (Z) and $E_{kin}$ developed by Waldvogel et al. (1978) with a calibration based on nine and twelve hail cells for buildings and cars, respectively. The obtained results also led to the development of the first probabilistic hail model for Europe, which was later purchased by "Risk Management Solutions" (Miller, 2007). Furthermore, Schmidberger (2018) developed a hail damage model for parts of Germany with the primary purpose to serve as a risk assessment tool. It derives hail tracks from radar data and hail sizes from the ESWD. The damage model considers separate impact functions for different building types, where each function is split into damages at the building exterior and a value for the roof-penetrating damages. Expected damages with high return periods are modelled with a stochastic simulation of 10'000 years with hail cell tracks and hailstone sizes. Similarly, Yin et al. (2007) developed an event-based hail risk model for cars in the US, based exclusively on bias-corrected hail reports. Hail losses are estimated by simulating 10'000 years of hail events with a Monte Carlo approach, and using three separate impact function for dent repair labour cost, parts replacement, and depreciation costs. Lastly, Ackermann et al. (2023) use a deep neural network to successfully estimate relative hail damages to buildings in Australia from S-band radar data and environmental parameters from reanalysis data.

While the existing models allow for a static, probabilistic hail risk assessment, they are not open-source and lack the capability to provide real-time hail impact estimates. This paper explores the potential of existing radar-based hail intensity measures to model damages to buildings and cars and is structured as follows: First, the radar, hail damage, and exposure data from Switzerland are introduced in Section 2. Leveraging 20 years of geolocated per-building damage data, we conduct a detailed verification of available radar products at 1 km resolution (Sect. 3). Within the open-source risk modelling framework CLIMADA (Aznar-Siguan and Bresch, 2019; Aznar-Siguan et al., 2023), we develop a hail damage model (Sect. 4) with focus on real-time hail damage assessments, following the concept of risk as used by, e.g., the IPCC (Pörtner et al., 2022), namely the combination of hazard, exposure and vulnerability. A detailed model evaluation (Sect. 5) highlights the skill and the limita-

tions of the MESHS-based hail damage model for buildings and cars, also in comparison with other radar variables. Section 6 discusses the results and the model applicability, and Sect. 7 summarizes the paper and highlights the key conclusions.

## 2 Data

To build a hail damage model, three datasets are required: hazard, informing about occurrence and intensity of hail (Sect. 2.1); exposure, informing about location and value of assets (Sect. 2.2; 2.3); and damage, informing about vulnerability of assets to hail (Sect. 2.2; 2.3). The datasets are described in the following. All hazard data in daily resolution comprise sub-daily information from 6 UTC to 6 UTC the following day. This threshold of 6 UTC marks the daily minimum of hail activity in the domain (Nisi et al., 2016; Schroeer et al., 2023) and so allows defining physically consistent hail events by minimizing the splitting of hail events over two calendar days.

### 2.1 Radar data

The five MeteoSwiss radars scan 20 elevation sweeps every five minutes (Fig. 1; Germann et al., 2015). Here, we use five different radar products: MESHS, reflectivity, VIL, and $E_{kin}$ are used as hazard variables and POH is used for the pre-processing of damage data. Both POH and MESHS are based on the algorithms developed by Waldvogel et al. (1979) and extended by Witt et al. (1998). POH is calculated from the relation between the 45 dBZ contour height of radar reflectivity and the freezing level derived from the operational weather model COSMO (Baldauf et al., 2011). It thus represents an estimate for the zone where hail may grow by riming in deep convective storms (Betschart and Hering, 2012). MeteoSwiss is using the polynomial fit by Foote et al. (2005) to convert the height difference to a probability of hail (0-100%). Similarly, MESHS is calculated as the height difference between the 50 dBZ contour and the freezing level, following the approach of Treloar (1998) and its first operational implementation by Joe et al. (2004) during the Sydney 2000 Forecast Demonstration project. In contrast to POH, MESHS is an estimate of the maximum hailstone size for values from 2 cm and above. While the algorithm has no upper limit, uncertainties are high particularly above 6 cm because of verification data limitations. Exact formulas and visualizations for the MeteoSwiss implementation of the POH and MESHS algorithms are provided in Trefalt et al. (2023).

The second hazard variable used in this study, $E_{kin}$, can also be estimated from radar reflectivity (e.g. Waldvogel et al., 1978) and has been used to model hail damages to cars and buildings by Hohl et al. (2002a). However, the method has large uncertainties and was originally calibrated with observations from an S-band radar. Cecchini et al. (2022) provide updated $E_{kin}$ estimates for C- and S-band radars separately, based on the same dependence to radar reflectivity but with adjusted coefficients. We here use the C-band specific coefficients of Cecchini et al. (2022), but follow the approach of Hohl et al. (2002a) by integrating over time and using a low-level Constant Altitude Plan Position Indicator (CAPPI) in order to compare our model with previous hail damage modelling approaches. In a first step, kinetic energy flux ($\dot{E}$, in $\mathrm{J\,m^{-2}\,s^{-1}}$) is calculated as:

$$\log(\dot{E}) = -6.72 + 0.114Z \tag{1}$$

where $Z$ is the radar reflectivity in dBZ at a 2 km CAPPI. In a second step, daily integrated values of $E_{kin}$ are calculated as sum of $\dot{E}$ over all 5 min time steps of a day (6–6 UTC).

The vertical maximum reflectivity, which is directly measured by the radar, is used as a third hazard variable, as it has previously also been used as a proxy for hail intensity (e.g. Mason, 1971; Kunz and Puskeiler, 2010). Lastly, the VIL is quantified from vertically integrated radar reflectivity, which is converted into liquid water equivalent according to Greene and Clark (1972).

    Only POH and MESHS are available in sufficient quality already from 2002-2012 due to extensive re-processing by Trefalt
et al. (2023). Reflectivity, $E_{kin}$, and VIL are used from 2013, where dual-polarisation radars allow for an automatic clutter filtering of sufficient quality without re-processing (Germann et al., 2015).

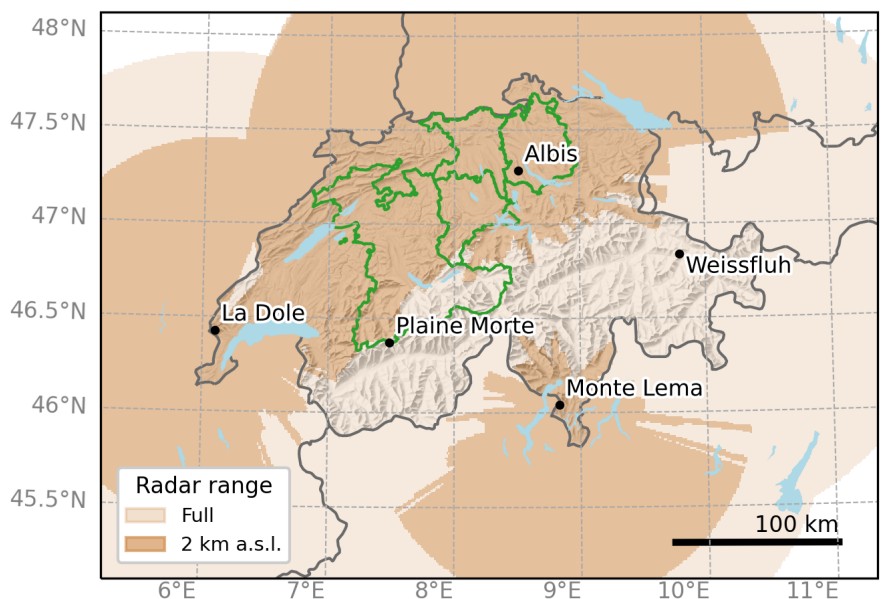

**Figure 1.** Radar locations in Switzerland with the full scanning range of 160 km (light brown) and the reduced scanning range at 2 km a.s.l. relevant for $E_{kin}$ (dark brown). The four cantons (from southwest to northeast: Berne, Lucerne, Aargau, Zurich) with available building damage data are outlined in green.

## 2.2 Building Data

Exposure and damage data from four cantonal building insurances in Switzerland are used: GVZ (Zurich), GVL (Lucerne), AGV (Aargau), GVB (Berne). These cantons contain 43% of the Swiss population. Building-level damage reports are available from 2002-2021 and exposure data consist of the complete buildings stock (>99% insurance fraction) in 2021. Since spatial coordinates were not available from some of the insurances, for Lucerne and Aargau addresses were geocoded using the OpenStreetMap API Nominatim (Haklay and Weber, 2008). This introduces some additional uncertainty as, e.g. buildings with an unknown street number are assumed to be located in the center of their street. Furthermore, few addresses (<0.5%) were geocoded with locations outside the cantonal border. Since cantonal building insurances only insure property within their canton these buildings must be incorrectly geocoded and were removed from the exposure data. In total, the dataset contains the spatial coordinates, building value (in CHF), and year of construction of 989'000 buildings with the reference year 2021. The buildings are distributed over 89% of all 1 km gridcells within the four cantons, the remaining 11% containing no building (Fig. 2a). Out of all gridcells with at least one building, 75% contain ten or more buildings. Average building values range from mostly below 1 million CHF in rural regions to over 10 million CHF in the cities of Zurich, Berne, and Lucerne, where land prices are high and large multistory buildings are common (Fig. 2b).

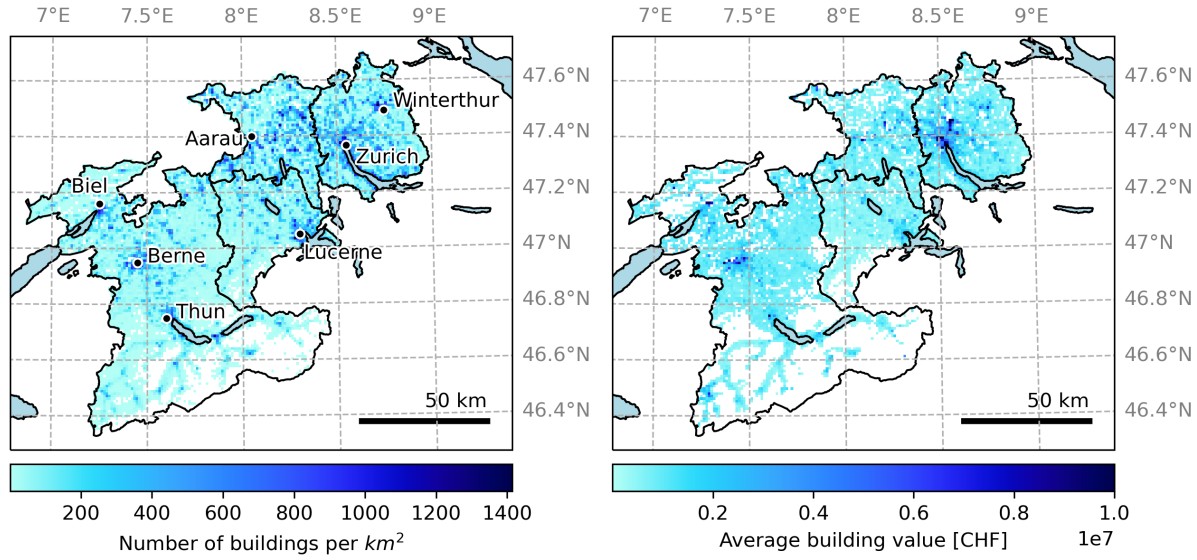

**Figure 2. (a)** Number of buildings per square kilometre in the four Swiss cantons with available data. **(b)** Average building value (for all $1\,\mathrm{km}^2$ gridcells with >10 buildings).

### 2.2.1 Building damages

The building damage data include 250'000 reports with spatial coordinates, date, total damage in CHF (incl. deductible), and a building ID to match it with the exposure data. To ensure consistent damage and exposure data, our dataset does not contain damage reports of buildings that are not part of the 2021 portfolio (i.e. torn-down buildings). The total claim volume over the considered 20-year period is 1.39 billion CHF distributed over 806 days, of which 90% (1.25 billion CHF) is caused by the strongest 30 hail days, with a single event on 28 June 2021 causing 35% (483 million CHF).

Some pre-processing of the damage data is required to use them for damage modelling. First, 38'787 damage claims report damages of zero and are removed, leaving 212'026 valid damage claims. Entries with zero represent claims that were not accepted as valid hail damages by the insurance. Second, all damage claims are indexed to the year 2021 using a canton-specific building cost index provided by the four building insurances. Please note that an accurate temporal reporting of hail damages is not always guaranteed, in particular for damages that occur at night or, in fewer cases, when residents are not at home during a hail storm.

Thus, damage data are pre-processed using a plausibility filter to correct for obvious misattributions. Firstly, areas with plausible hail for each day are derived by requiring a POH of 10% with a 5 km buffer to account for the 2–4 km wind drift of hailstones (Hohl et al., 2002b; Barras et al., 2019). 10% POH represents a conservative threshold as damaging hail (>2cm) is expected for a POH of >80% (Saltikoff et al., 2010), but Nisi et al. (2016) note that soft hail or graupel in Switzerland can be detected in lower POH values of 20–50%. If within ±2 days of a hail damage report, no plausible hail is observed, the damage report is removed (3'460 of 212'026 cases). If only one day has plausible hail within ±2 days at the location of a hail damage report, the report is assigned to this day. If hail is plausible on more than one day, the date is only changed if another day has at least a 50% higher POH than at the originally reported date. The correction is applied to 57'432 of 212'026 damage reports, of which 92.5% were corrected to the previous day, indicating that damages were reported on the day after a hail storm. As hail activity often continues into the night and over midnight, this shifting of damage claims aligns the claims with consistent hail events (6–6UTC; Sect. 2). While the chosen POH-based pre-processing is not fully independent of other radar-based hazard data, it is required when working with insurance claim data and thresholds are chosen conservatively compared to other studies (e.g. assigned hail damage claims within ±30 days of the reported date in Warren et al., 2020).

The pre-processed building damage data reveal almost log-normally distributed values averaging at 6'650 CHF and ranging from 395 to 21'900 CHF (5-95% quantile), with 1187 claims above 100'000 CHF (Fig. 3). The peaks at CHF 500 and CHF 5000 stem from cases where the insurance paid a fixed amount rather than the actual repair costs. As expected, claim values increase with the value of the affected building, but not linearly (parallel to the 1:1 line in Fig. 3). On average, the damage is proportional to the building value to a power of 0.27, indicating a decrease in relative damage with increasing building values. This decrease reflects the fact that while the value of a building is approximately proportional to its volume, hail damages are proportional to the area (and value) of exposed vulnerable building parts (blinds, roof, and facade) which scale roughly with the building surface area. Within this study, we use both, the absolute building value and a scaled version which approximates the value of exposed vulnerable buildings parts as being proportional to the building value to a power of 0.27.

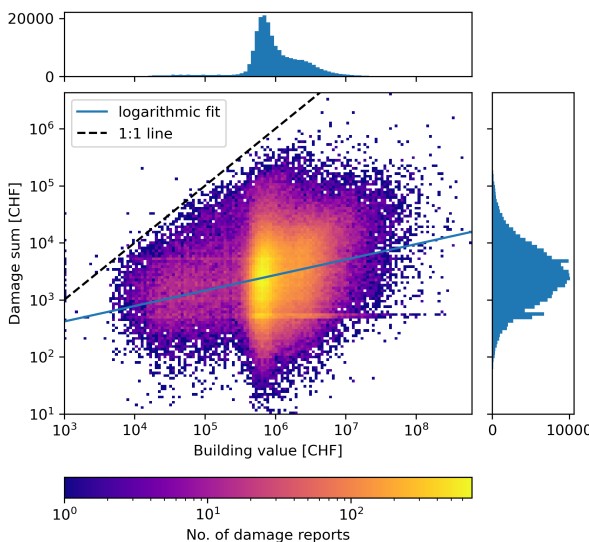

**Figure 3.** Density plot of building values vs. reported damage for the 212'026 valid damage claims in the considered cantons from 2002–2021. A linear fit (in the logarithmic space) is shown in blue, and the 1:1 line in black.

## 2.3 Car data

Car damage and exposure data for the model calibration are provided by a private insurance company for the years 2017-2021. Each year, over 500'000 cars are insured over all Swiss cantons, with a total value of over 10 billion CHF. Reported claims vary strongly from year to year, with a total of over 50'000 claims over the five-year period 2017–2021. Due to conditions of the data provider, per-event reported car damages are only explicitly shown as normalized values.

In contrast to buildings, the exact location of a car (and whether it is directly exposed to hail or covered for protection) at any given time is unknown. Thus, the location of each vehicle is inferred from the municipality of the registered address of its most frequent driver (mostly the owner). Since we are interested in modelling the portfolio losses, rather than an individual car, the location of each car is assumed to be a random point within the municipality. The average area per Swiss municipality is 12.9 km$^2$ with larger areas in mountainous regions and smaller values in cities where the density of cars is highest.

Given the uncertainty in the spatial coordinates of a vehicle, the POH-based plausibility filtering must be adjusted, compared to the building damages. Rather than defining if a claim is in an area with plausible hail, for each day we create a likelihood function that a car registered at a given coordinate could have driven into a hail storm. Based on the average Swiss municipality area of 12.9 km$^2$ and the average car ride distance in Switzerland (14,8 km with 63% of all rides <10 km; Biedermann, 2023) we select a 50 km radius which covers most cars, as visually determined by analyzing reported car damage claims during large events with known hail extent. We assume the likelihood to be proportional to the number of gridcells with a POH>10% (see Sect. 2.2.1) within the 50 km radius. As damages are particularly often reported the day after a hail event (due to nocturnal hailstorms), the date is changed if the likelihood at the location of a report is over twice as large as on the reported date. For 2

days before or 1–2 days after, the date is only changed if the likelihood is >10 times higher than on the reported date (usually cases where no POH at all is detected on the date of the report). This approach removes 1.7% of all claims where no POH within 50 km is reported, and moves 9.5% to the previous day and 4.9% to $\pm 2$ days. Of course, individual cars drive further than 50 km and a few correct damage reports will be shifted to a different date, which is unavoidable when working with spatio-temporally imprecise damage reports.

## 3 Evaluation of radar-based hail intensity measures

Before calibrating a hail damage model, we evaluate the available radar-based hail intensity metrics using the geolocated building damage and exposure data. The focus of this evaluation lies on the occurrence of damaging hail and its spatial match with radar data rather than its intensity, which will only be used in the model calibration in Sect. 4. Since exact spatial coordinates of reported hail damages are required to verify occurrences of damaging hail, only data for buildings and not cars are used in this section. Both damage and hazard data are classified to binary values. Each $1\,\text{km}^2$ gridcell with 10 or more buildings is classified as hail damage (yes/no) on each date. Different thresholds ranging from 1 to 100 buildings per gridcell were tested. A threshold of 10 was chosen as it retains 75% of all gridcells, while avoiding conclusions about the hail occurrence based on few buildings only. Hail intensity metrics are divided into gridcells with damaging hail expected (yes/no) by a threshold which is varied over the respective intensity range (e.g., from 20 to 80 mm for MESHS). For each variable and intensity threshold, we so obtain hits (H), false alarms (FA), misses (M), and correct negatives (CN) to create a contingency table (Table 1). From these fractions, the skill metrics probability of detection (POD), false alarm ratio (FAR), and Heidke-skill-score (HSS; Wilks, 2019) are calculated as shown below. The HSS is suitable to quantify forecast skill of rare events like hail storms (Doswell et al., 1990; Wilks, 2019) and has previously been used to verify radar-based hail observations (e.g., Kunz and Kugel, 2015; Warren et al., 2020).

$$POD = \frac{H}{H+M} \tag{2}$$

ranging from 0 (no skill) to 1 (perfect model).

$$FAR = \frac{FA}{FA+H} \tag{3}$$

ranging from 0 (perfect model) to 1 (no skill).

$$HSS = \frac{2 \cdot (H \cdot CN - FA \cdot M)}{(H+M)(M+CN) + (H+FA)(FA+CN)} \tag{4}$$

ranging from -1 to 1 (perfect model), with positive values denoting better and negative values lower skill than a random guess.

**Table 1.** Standard 2x2 contingency table for dichotomous events, used for validation of damaging hail occurrence.

|  | damage observed | no damage observed |
|---|---|---|
| hail predicted | Hits (H) | False alarm (FA) |
| no hail predicted | Misses (M) | Correct Negatives (CN) |

Figure 4a–c shows the skill scores for MESHS, maximum reflectivity, and $E_{kin}$ over their whole intensity range. As expected, both POD and FAR decrease with an increasing intensity threshold for each variable. Thus, there is a trade-off between detecting most hail events at a low threshold and obtaining few false alarms at high thresholds. The HSS considers both of

these effects and peaks at intermediate thresholds. Note that an evaluation on a 1 km gridcell level yields more conservative skill estimates relative to studies with less accurate damage data (e.g. on community level; Skripniková and Řezáčová, 2014; Kunz and Kugel, 2015; Puskeiler et al., 2016; Nisi et al., 2016). Because a part of the increased skill at lower resolution could be explained by hail drift, we additionally calculate skill scores with explicit hail drift considered. Hazard data for each day are artificially shifted to maximize the correlation to the percentage of affected buildings in all gridcells with 10 or more buildings. As the highest damages are mostly concentrated in one hail streak, we only consider one hail drift vector per day with a maximum length of 4km accounting for the 2–4 km wind drift of hailstones (Hohl et al., 2002b; Barras et al., 2019). Skill scores obtained from the shifted hazard data are shown for comparison in Fig. 4 and consistently show a marginal improvement over the original skill scores. Given the minimal improvement, but inability to consider damage data-dependent hail drift in real-time applications, we focus on the non-shifted hazard data for model development.

At the minimum MESHS value of $20\,\mathrm{mm}$, the POD reaches 60% for an exact spatial match, indicating that 60% of all gridcells with a hail damage report lie within a MESHS footprint. When analyzing individual claims (not shown), detection probabilities of 88% are reached, indicating that missed events are often cells with few claims, while cells with many claims are more likely to be within a MESHS footprint. The gridcell-wise FAR decreases almost linearly with increasing MESHS from over 80% at $20\,\mathrm{mm}$ to 50% at $80\,\mathrm{mm}$. Thus, even at extreme MESHS values only one in two gridcells with >10 buildings contains a damage claim and in 20% of cases there is not a single damage claim within 4 km (not shown). Overall, the skill of MESHS is highest between 20 and $50\,\mathrm{mm}$ with a HSS of >0.2 and a peak of 0.3 at MESHS equal to $35\,\mathrm{mm}$.

In contrast to MESHS, maximum reflectivity is defined on a continuous scale without minimum value and the shown range starts at 40 dBZ, where both POD and FAR reach almost 100%. As POD and FAR decrease, maximum reflectivity starts showing some predictive skill with the highest HSS values of over 0.2 in a window between 55 and 65 dBZ. The enhanced HSS above 50 dBZ, which is used as threshold in MESHS, indicates that there is additional information regarding the occurrence of damaging hail in high reflectivity values up to 65 dBZ.

$E_{kin}$ values are distributed in a long-tailed distribution with few high values and the majority below $200\,\mathrm{J m^{-2}}$. Thus, the POD decreases rapidly while the FAR decreases linearly with increasing $E_{kin}$ reaching 60% at $1000\,\mathrm{J m^{-2}}$, comparable to the FAR of MESHS at $80\,\mathrm{mm}$. However, few gridpoints show such high $E_{kin}$ values leading to low POD and, accordingly, HSS. The highest predictive skill is achieved between 200 and $400\,\mathrm{J m^{-2}}$ with a HSS of 0.2. Lastly, VIL (not shown) has lower HSS and higher FAR than the three variables shown in Fig. 4 indicating a weak distinction of hail and non-hail events even at high values.

A maximum HSS of 0.3 over all variables shows that there is no radar variable threshold that can reliably identify all areas with damaging hail on a gridcell level. Spatial footprints in Fig. 4d–f and A1 exemplary show that within an intense hail streak nearly all gridcells consistently contain a damage claim, but no radar variable consistently distinguishes these hail streaks. The consistently observed damages within a hail streak and the minimal improvement in FAR with explicit hail drift indicate that the false alarms are largely related to limitations in radar-based hail proxies rather than variable building vulnerability or wind-driven hail drift. Of the three analyzed radar metrics, MESHS stands out as most promising with an extended range of relatively high HSS which offers good trade-off between POD and FAR. However, in particular the high FAR confirms that

MESHS cannot be interpreted as *actual* local hail size, but (as the name suggests) as *maximum expected* hail size, given the storm intensity. For a deeper discussion of MESHS and local hail size see Schroeer et al. (2023). Since no radar-based proxy corresponds accurately to hail size or energy, the calibration in the following section is based on empirical hazard-vs-damage relationships, rather than physical considerations of the vulnerability of buildings materials to hail sizes (e.g. Stucki and Egli, 2007; Macdonald and Stack, 2021).

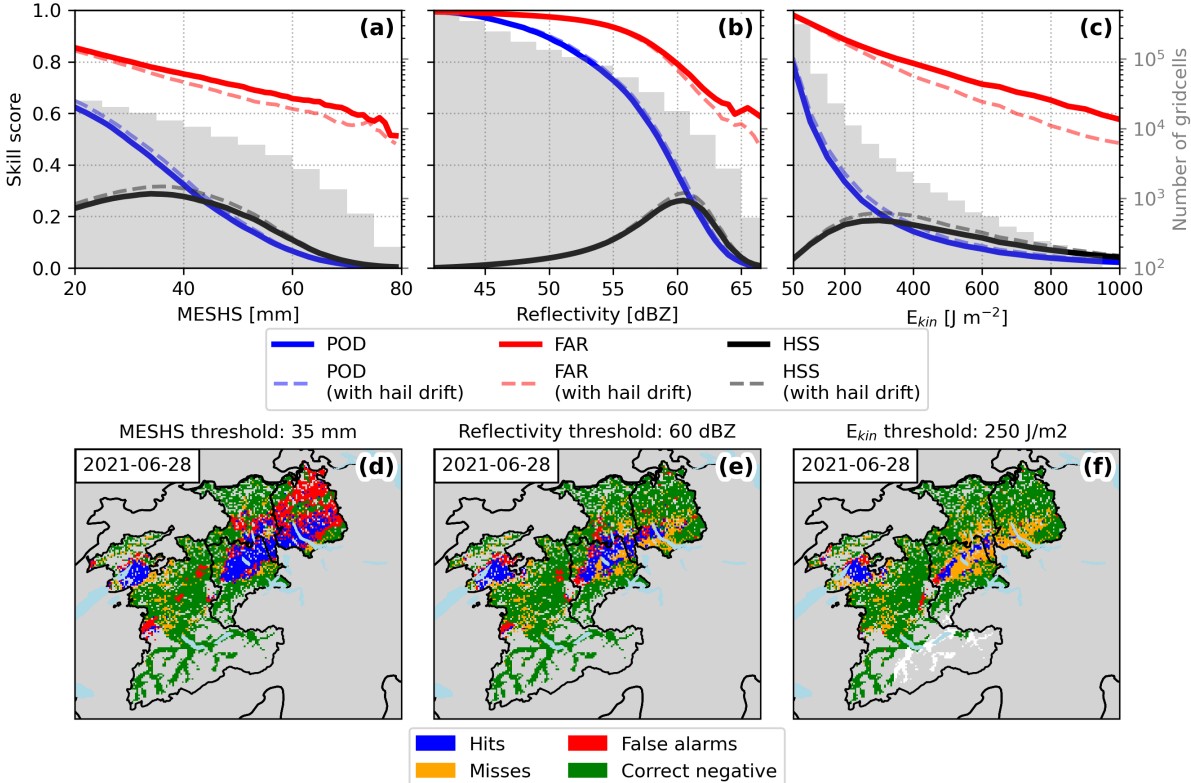

**Figure 4. (a,b,c)** MESHS, maximum reflectivity, and $E_{kin}$ verification statistics probability of detection (POD), false alarm ratio (FAR), and Heidke-skill-score (HSS) calculated per gridcell for all 1km cells with >10 buildings. Dashed lines indicate the same skill scores with post-processed hazard data, which explicitly captures unidirectional hail drift by maximizing the correlation to observed relative damages (refer to text for details). Grey bars show the number of gridcells with given hazard intensity over the whole time period with a logarithmic scale. **(d,e,f)** Visualization of contingency table variables per gridcell for a single event (28 June 2021) and selected hazard thresholds corresponding to the maximum HSS. Grey areas do not contain sufficient exposure data (<10 buildings) and white areas mark cells with missing hazard data.

## 4 Model calibration

We build a hail damage model for each of the introduced radar variables MESHS, maximum reflectivity, and $E_{kin}$. Based on the analysis in the previous section, we focus on the MESHS-based model. However, model parameters and skill scores are provided for all hazard variables to accommodate model users without access to MESHS data. Following the concept of risk as a combination of hazard, exposure, and vulnerability (IPCC; Pörtner et al., 2022), the model estimates hail damages by translating a hazard intensity (e.g. MESHS) to a relative damage with an impact function. This relative damage is translated to absolute values (e.g. number of damaged buildings or damage sum in CHF) by multiplication with the exposure data in a spatially explicit fashion (Aznar-Siguan and Bresch, 2019).

Impact functions for different natural hazards are typically calibrated by assuming a parameterized impact function (e.g. sigmoid function) and minimize a loss function between modelled and observed per-event damages (e.g. Schwierz et al., 2010; Eberenz et al., 2021; Lüthi et al., 2021). In this study, the point-based building damage data allow for a spatially explicit calibration which can additionally inform about the shape of the impact function and provide detailed uncertainty estimates. The approach consists of the following steps:

1. Select all days with non-zero hazard intensity and/or a damage report from the considered time period of April–September (i.e. convective season) 2002-2021 (806 days for MESHS and the building damage reports).

2. Assign the local hazard intensity to each damage report and exposure point in the dataset (for events before 2021 the buildings that were not yet built are removed).

3. Aggregate exposure and impact variables (number and value of exposed buildings/cars, number of claims, damage claim sum) to hazard intensity levels (MESHS values from 20-100 mm in 1 mm intervals). Note that for high hazard intensities data points become increasingly sparse.

4. For each hazard intensity level, calculate the average percent of assets affected (PAA) and mean damage ratio (MDR):

$$PAA = \frac{N_{dmg}}{N_{exp}}, \quad MDR = \frac{V_{dmg}}{V_{exp}} \tag{5}$$

where $N_{dmg}$ is the number of damaged assets, $N_{exp}$ is the number of exposed assets, $V_{dmg}$ the damage costs, and $V_{exp}$ the monetary value of exposed assets. For high hazard intensities purely empirical PAA and MDR become more uncertain and eventually unstable due to under-sampling of the underlying distribution (see Sect. 4.1).

5. Calculate two types of impact functions from the raw impact data per hazard intensity. First, calculate a monotonically increasing smooth-empirical fit by using a running mean (10 mm running mean for MESHS). Given the aforementioned under-sampling, above a certain hazard threshold, the smooth-empirical impact function is given as constant by the average PAA/MDR for all values above this threshold (60 mm for MESHS). Secondly, fit a sigmoid-type function as proposed by Emanuel (2011) by minimizing the root-mean-squared error (RMSE) between the function and the purely empirical PAA or MDR:

$$f = \frac{v_n^3}{1+v_n^3} \cdot S, \quad v_n = \frac{MAX[(V-V_{thresh}),0]}{V_{half}-V_{thresh}} \tag{6}$$

where $f$ is the relative impact (PAA or MDR), $S$ the scale parameter (i.e. maximum relative loss), $V$ is the hazard intensity (here MESHS), $V_{thresh}$ the hazard intensity below which no damage occurs, and $V_{half}$ the hazard intensity at which the relative loss is half of its maximum (i.e. half of the scale parameter $S$). The RMSE is minimized using the Nelder-Mead algorithm (Gao and Han, 2012) with parameter bounds informed from the empirical PAA or MDR. For MESHS, $V_{thresh}$ ranges from 0-20 mm, $V_{half}$ from 20 mm to the maximum considered MESHS value (100 mm), and $S$ is constrained within two orders of magnitude of the maximum PAA or MDR of the empirical function (for MESHS-based building damages: PAA: 0.6%–60%, MDR: 0.005%–0.5%).

6. To quantify effects of sampling uncertainty on the impact functions, a bootstrapping is used: Steps 3–5 are repeated for 1000 subsamples, each including 806 days. The subsamples are obtained by randomly selecting 806 days with replacement from all 806 days identified in (1). Note that Appendix D additionally provides a 5-fold cross validation (CV) analysis which highlights that (a) impact functions for each CV split are well captured with the bootstrapping approach, and (b) that the model skill is equivalent on independent verification data.

## 4.1  Calibrated impact functions for buildings

Before showing the calibrated impact function (step 5 of the calibration procedure), we shortly consider the observed gridcell-wise variability in PAA and MDR (step 3). For all MESHS intervals shown in Fig. 5, many gridcells with buildings exposed to high MESHS values still have zero reported damages (false alarms) and, thus, zero PAA and MDR. The fraction of these gridcells is equivalent to the FAR in Fig. 4 and represents grid cells with buildings where hail is expected according to the radar, but no damage occurred. Most of these false alarms as well as low damages occur for MESHS below 40 mm (Fig. 5c,d) while high relative damage can be predominantly attributed to MESHS values above 40 mm (e.g. PAA >20% or MDR >0.1%; Fig. 5c,d) Note that this increase is more pronounced for the PAA (Fig. 5c) because the MDR is additionally dependent on the building value and, thus, increasingly variable when considering fewer gridcells. The transitioning distribution of PAA and MDR as MESHS increases results in ascending average values of PAA and MDR (step 4) which ultimately determine the shape of the vulnerability curve (step 5). This, in turn, means that the calibrated impact function does reproduce the observed gridcell-wise variability of PAA and MDR. Instead, it assigns the same relative damage (PAA or MDR) to each gridcell with the same MESHS value, corresponding to the average of the distribution in Fig. 5a,b.

The resulting smooth-empirical impact functions for buildings (Fig. 6) start at almost zero at 20 mm MESHS, reaching 5% for PAA and 0.05% for MDR at 60 mm with a bootstrap 90% confidence interval (CI) between 3-7% for PAA and 0.02-0.09% for MDR. The larger relative CI for MDR indicates higher sensitivity in hail damage sums towards individual events compared to the number of affected buildings. The bootstrap CIs accurately quantify the sampling uncertainty but do not reflect the variability of gridcell-wise PAA and MDR given a certain MESHS value (discussed earlier; Fig. 5a,b). Above 60 mm MESHS, fewer damage reports (red bars in Fig. 6) and fewer affected gridcells (6304 from 40-60mm vs. 1435 from 60-80mm) remain.

Hence, and given the long-tailed distribution of PAA and MDR values (Fig. 5a,b), we refrain from continuing the smooth-empirical impact function for these MESHS values.

To provide impact function estimates for higher values, we use a sigmoidal fit (Emanuel, 2011), which is fitted to all data (including values above 60 mm), but weighted by the number of samples (see Sect. 4). This approach avoids overfitting to high MESHS values with few data points and provides vulnerability information for the complete MESHS range, with increasing
uncertainty for high values. The sigmoidal fit (Fig. 6, in black) agrees well with the empirical PAA and extends it to reach 5–10% at 80 mm MESHS. The sigmoidal function corresponds to Eq. (6) with the parameters from Table 2. For the MDR, the empirical fit exhibits a steeper increase between 50 and 60 mm, which cannot be fully captured by the chosen sigmoidal fit. While the increase in MDR up to 50mm is largely driven by increasing PAA, the sharp increase above 50mm MESHS occurs due to a strong increase in mean severity, partly caused by extreme (roof-penetrating) damages. Mean severity quantifies the
damage ratio, given that a damage claim was made. Thus, the mean severity multiplied with the PAA corresponds to the MDR, assuming that the average value of damaged buildings corresponds to the average value of all buildings. However, regarding the final model skill the difference between the empirical fit and the sigmoidal fit is small (Appendix C) as hazard data uncertainties dominate (Sect. 5). For the subsequent damage modelling the sigmoidal fit is used. Note that impact functions were also derived for the hazard variables $E_{kin}$, maximum reflectivity, and VIL (see Table 2 for the fitted parameters and Sect. 5
for their discussion).

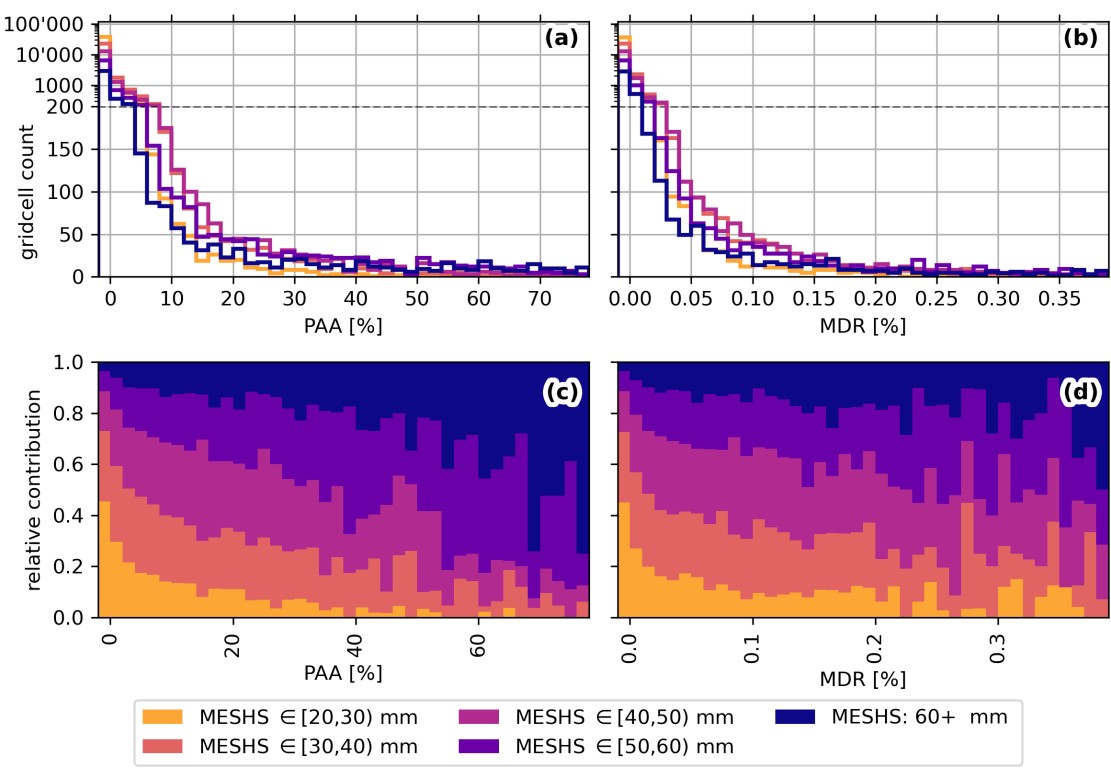

**Figure 5. (a,b)** Distribution of PAA and MDR values for all $1\,\mathrm{km}^2$ gridcells with 10 or more buildings, split by observed MESHS values. **(c,d)** Corresponding relative contribution of each MESHS class. Note that the first bin comprises gridcells with a PAA and MDR value of zero and contains over 50% of all gridcells.

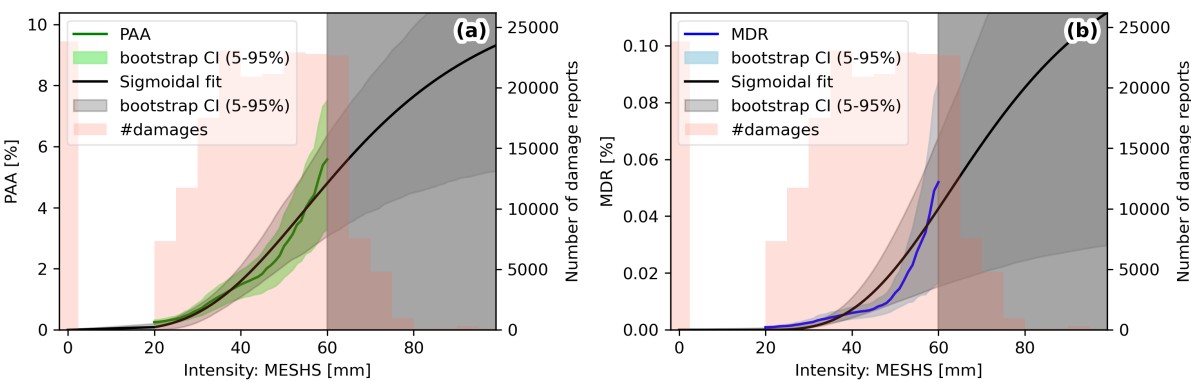

**Figure 6. (a)** Impact function for the number of damaged buildings, calibrated with data from 2002-2021. The solid green line indicates the 10 mm moving average for the empirical percent of assets affected (PAA) and the green shading its 5-95% bootstrap confidence interval. In black, a sigmoidal fit (Emanuel, 2011) is shown, and pink bars indicate the number of damage reports per MESHS intensity. The grey shaded area indicates MESHS values without sufficient data points for a meaningful empirical fit. **(b)** same as (a), but for damage sums in CHF, with the empirical mean damage ratio (MDR) in blue.

## 4.2 Calibrated impact functions for cars

Regarding car damage modelling, the inaccurate spatial coordinates of each individual car constitute an additional challenge for the impact function calibration. As the exposure data contains a large portfolio and the probability distribution of the actual car location decreases with increasing distance to the assumed location in its municipality (Sect. 2.3), an impact function can still be calibrated empirically. As expected, many car damages (42% vs. 12% for buildings) seemingly occur at a MESHS value of zero (Fig. 7), many of which likely were in an area with MESHS when the hail damage occurred. As the calibration is conducted with the complete car portfolio, the resulting impact function implicitly considers the probability of cars being sheltered (e.g., in a garage) or spatially displaced. Thus, assuming similar vehicle statistics (e.g., average driving distance or ratio of sheltered cars) as in the calibration dataset, the impact functions can be directly applied to a car portfolio with the location given by the address of each vehicle's most frequent driver. Empirical impact functions (Fig. 7) show a robust increase with MESHS from a PAA of 0.2% to 2.75% at 60 mm MESHS and a MDR of 0.02% to 0.25%. For both PAA and MDR, the sigmoidal fit follows the empirical function closely, with slightly lower vulnerability at MESHS below 30 mm. The lower PAA compared to building damages is consistent with many cars being covered in a garage, preventing hail damage. In contrast, the high MDR compared to buildings is due to lower exposure values for cars, relative to the inflicted damages by hail. As for the buildings, the simgoidal fit (Table 2) is used for subsequent damage modelling.

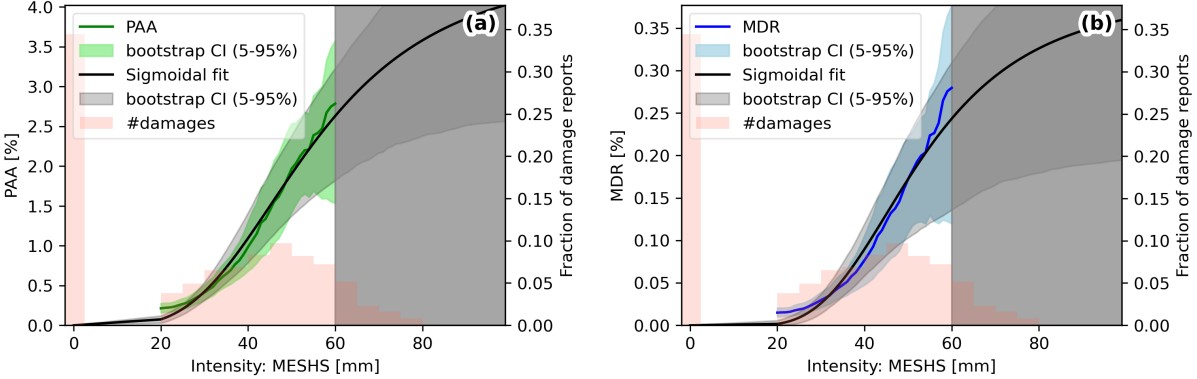

**Figure 7.** As Fig. 6, but for car damages, calibrated with data from 2017-2021. Due to conditions of the data provider, pink bars show the fraction of damage reports per MESHS intensity, rather than absolute values.

**Table 2.** Parameters of best-fit sigmoidal impact function for building and car damages as described in Sect 4.

| Exposure | Hazard | $V_{thresh}$ | $V_{half}$ | scale |
|---|---|---|---|---|
| (a) No. of damage claims (PAA) | | | | |
| buildings | MESHS [mm] | 8.6 | 66.9 | 0.118 |
| buildings | Max. reflectivity [dBZ] | 53.3 | 74.5 | 0.318 |
| buildings | $E_{kin}$ [J m$^{-2}$] | 0 | 465 | 0.037 |
| buildings | VIL [g m$^{-2}$] | 7.2 | 42.5 | 0.072 |
| cars | MESHS [mm] | 8.1 | 55.0 | 0.046 |
| (b) Damages (MDR) | | | | |
| buildings | MESHS [mm] | 20 | 73.9 | 1.47e-3 |
| buildings | Max. reflectivity [dBZ] | 52.1 | 74.5 | 1.80e-3 |
| buildings | $E_{kin}$ [J m$^{-2}$] | 0 | 480 | 2.50e-4 |
| buildings | VIL [g m$^{-2}$] | 15.8 | 29.3 | 1.80e-3 |
| cars | MESHS [mm] | 13.7 | 53.3 | 3.96e-3 |

## 5 Model evaluation

### 5.1 Model evaluation metrics

The calibrated impact functions from the previous section allow estimating hail impacts to buildings and cars for each day with available hazard and exposure data. Here, we evaluate the model performance of deterministic hail impact estimates based on daily MESHS footprints from 2002-2021 and the available building exposure data. Both the number of damaged buildings and the total damage are compared to reported hail damage claims of each day. Given the hazard availability (Sect. 2), results for buildings are evaluated separately for the periods 2002-2012 (MESHS only) and 2013-2021 (all hazard variables), and results for cars are shown for the whole period with available damage data (2017-2021)

The model is primarily evaluated based on its ability to correctly estimate the total number and cost of hail damages over the four considered cantons. As the vulnerability of varies among individual buildings/cars and hail is a small-scale difficult-to-observe hazard (Martius et al., 2018), co-occurring with strong winds and precipitation which additionally influence damages, a large spread in hail damage costs is expected. To compare performance between the different model versions, we split per-event hail damages estimates into three categories. Hits are all events with modelled damages within one order of magnitude (OOM) of the reported values (green in Fig. 8), false alarms are events where the modelled damages are more than one OOM higher than reported damages (red in Fig. 8), and misses are days with modelled damages of at least one OOM lower than reported damages (orange in Fig. 8).

From these categories and considering all events above a minimum damage threshold, we derive the three model evaluation metrics: (1) model-probability of detection ($POD_M$) as one minus the fraction of misses, given observed damages above the threshold, (2) the model-false alarm ratio ($FAR_M$) as the fraction of false alarms given modelled damages above the threshold, and (3) the fraction of hits ($FH_M$) as the number of hits divided by all considered events (red, green, and orange in Fig. 8). For buildings the minimum damage threshold corresponds to 100 affected buildings or 100'000 CHF (modelled or observed) and for cars equivalent thresholds are chosen, but only normalized values are shown in Fig. 8c,d due to the conditions of the data provider (see Sect. 5.3 for details). The chosen thresholds ensure that all medium- to high-impact events are included, with the non-considered events (grey shaded in Fig. 8) constituting <1% of all damages and <5% of all claims.

For certain practical applications of the damage model, a skill assessment in terms of model-to-observed ratio may be more meaningful than the number of events with modelled damages within one OOM of the observed damages. Thus, for the best-performing hazard variable, MESHS, we additionally quantify how close within the correct OOM the observed damages and model estimates are.

### 5.2 Hail damages to buildings

Table 3 shows skill scores for hail impact model versions with different hazard and exposure data. A comparison to skill scores using the smooth empirical fit (Table C1) shows only minor differences and, thus, further results focus on the sigmoidal impact function. Compared to other hazard variables, MESHS performs best for both the number of damaged buildings and the total damage with a $FH_M$ of 91 and 77%, respectively. When also considering data before 2013, the $POD_M$ remains almost constant,

but significantly more false alarms (beige dots in the red shaded area of Fig. 8a,b) reduce the model skill. Many of these false alarms are days with multiple few-km-wide patches of positive MESHS values but few or no damage reports. After 2012, the fraction of false alarms reduces, which is likely explained by improved automatic pre-processing with dual-polarization radars (Sect. 2.1; Trefalt et al., 2023).

Similarly, the increased skill after 2012 is highlighted in the CV analysis in Appendix D, where all skill scores are consistently highest for the two CV splits after 2012. Crucially, the CV also shows that model skill on independent verification data nearly corresponds to the model trained on the complete time period (Table D1). In contrast, the chosen thresholds of the minimum considered damage per event have a large impact on skill scores. As evident from Fig. 9, both the $POD_M$ and $FH_M$ increase with increasing thresholds for minimum considered damage, while the $FAR_M$ remains stable. For example, if one is only interested in high-impact events (CHF 1 million and larger), model skill increases from $POD_M$ 80 to 94% and from $FH_M$ 77 to 83% for MESHS-based building damages. Focusing on observed extreme hail days from 2013–2021 (2002–2021), the largest 17 (33) events which make up 92% (91%) of all hail damages are modelled with the correct OOM. In comparison, the threshold for the minimum number of claims has a lower impact on the skill scores, as visible in Fig. 9a.

In order to quantify how close within the correct OOM hail damage estimates are, we additionally report the model-to-observed ratio. Considering all events after 2012 with more than 100 buildings affected, 50% of the MESHS-based model estimates are less than a factor of 2.8 off the reported number of damaged buildings and 75% less than a factor 5.1. Focusing on all events over 100'000 CHF building damage, 50% of the MESHS-based monetary hail damage estimates are within a factor of 4.2 of the reported damages and 75% within a factor of 8.7. As with the aforementioned primary model evaluation metrics, the skill in terms of modelled-to-observed ratio decreases for events before 2012 (beige dots in Fig. 8a,b), where more model estimates are further than one order of magnitude off the observed damages.

The other hazard variables $E_{kin}$, reflectivity, and VIL have a similar $POD_M$ compared to MESHS, but a higher $FAR_M$. Namely, there are more events where high damages would be expected from the hazard variable but are not observed, which also leads to an increased number of events which exceed the chosen thresholds of 100 damage claims or 100'000 CHF. Due to the significantly lower skill of other hazard variables, model skills for different exposure layers are only shown for models with MESHS as hazard. Considering scaled building exposure which approximates the value of the exposed building exterior (Sect. 2.2) does not change the model performance for aggregated damages substantially (Table 3). It does reduce overestimation of damages on individual expensive buildings, but as our model aims to estimate aggregated rather than building-level damages, there is little added value in using the scaled exposure.

## 5.3 Hail damages to cars

As with buildings, car damages are estimated with the calibrated impact function (Fig. 7) for the time period with available damage data (2017-2021). To compare model skill between building and car damages, appropriate thresholds for the minimum considered number of claims and damages must be selected. As the absolute exposure value differs, we choose thresholds that correspond to the same percentage of included damages. The chosen 100 buildings correspond to 95% of all damage claims

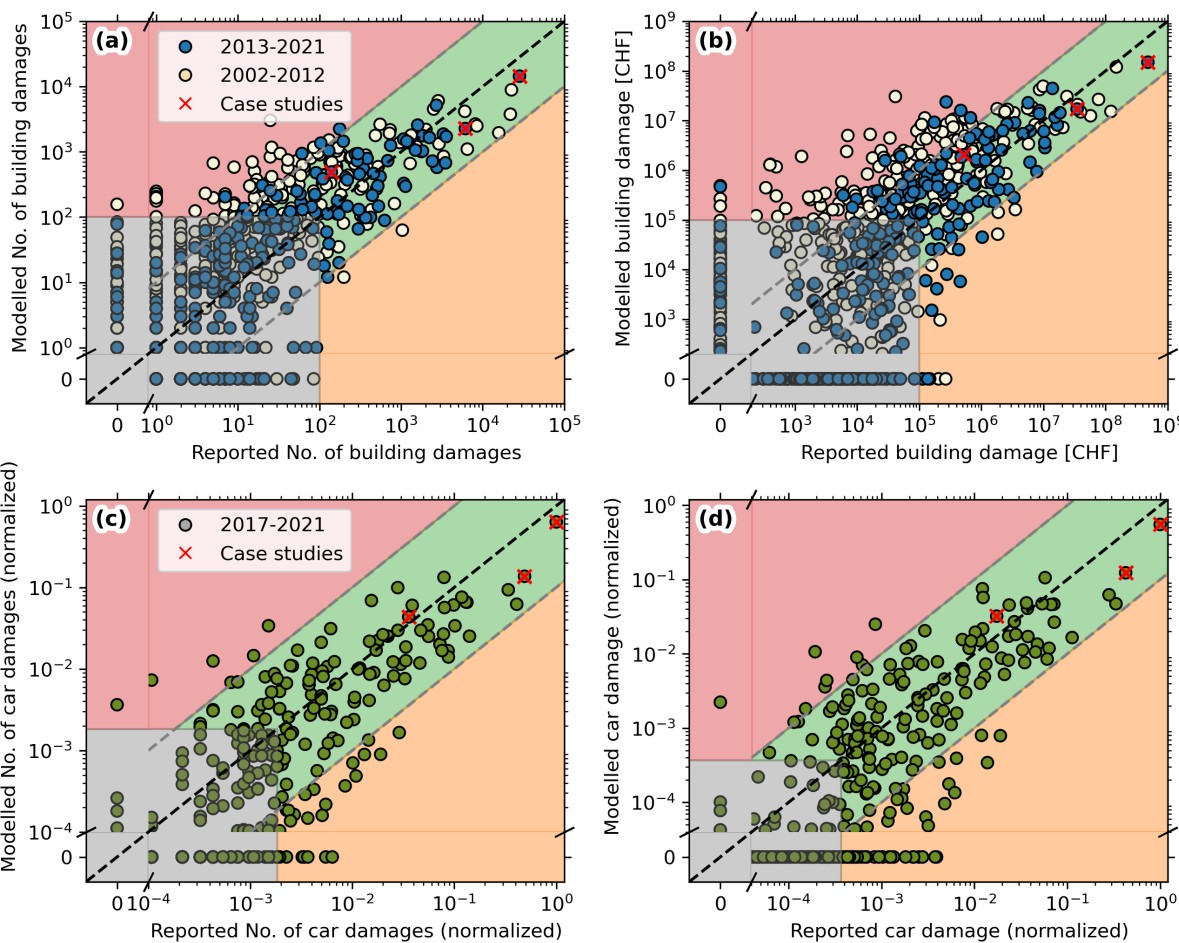

**Figure 8. (a)** Reported vs. modelled number of damaged buildings per hail event (1 day) for the years 2002-2012 (beige) and 2013-2021 (blue). Grey shaded are events with <100 affected buildings which are excluded for the calculation of the skill metrics. Background colours indicate hits (green), false alarms (red), and misses (orange). Case studies from Fig. 10 are marked with a red x. **(b)** Same as (a), but for total damages per event, with events <100'000 CHF damages excluded from the skill metrics calculation. **(c)** Reported vs. modelled number of car damages per event for the years 2017-2021 (green), normalized relative to the maximum impact event. Grey shaded are excluded events, with the threshold chosen to include the same percentage of the total claims as for buildings (99%). **(d)** Same as (c), for total car damages per event, with the threshold chosen to include the same percentage of total damages as for buildings (95%).

included, and 100'000 CHF correspond to 99% of the total damage sum included. We choose the equivalent percentage for cars, but only normalized values are shown in Fig. 8c,d due to the conditions of the data provider.

Figure 8 shows that the car damage model consistently predicts the correct order of magnitude for the 25 most impactful events, which contain over 80% of the total damage. However, skill metrics are lower than for buildings (Table 3) with a $POD_M$ and $FH_M$ of 74% for the number of damaged cars and 60% for total damages. The lower skill is explained by a high

number of missed events with zero modelled damages but observed damages above the chosen threshold, which relates to two main factors. Firstly, the assumed random location increases uncertainties particularly in the damage data pre-processing, which can lead to missed events. Secondly, the shorter considered time period contains fewer extreme events and, thus, a

425 certain percentage of included total damages contains more small-scale events. Furthermore, specifically for car damages the time of day is relevant due to increased traffic during rush hour, which is not considered in this study but may further improve damage estimates. As with buildings, there are events where missed damages in one location are compensated by over-predicted damages in another location where the hazard variable wrongly predicts intense hail (see next section). This compensation mechanism is more effective for larger spatial scales, which benefits skill scores for car damages that cover all

430 Swiss cantons.

As expected from Fig 8b,c the model-to-observed ratio tends to be larger for cars than buildings. The 50% quantile is at a factor of 3.8 for the number of damaged cars and 5.4 for the monetary damages, while the 75% quantiles are beyond one OOM. However, when only focusing on the 25 strongest events (containing 80% of all damages) the 50% quantile for both number of claims and monetary damages is below a factor 3 and no event exceeds a factor 8, indicating improved model performance.

**Table 3.** Skill scores for calculating hail damages to buildings (cars) with different variable combinations for the period 2013-2021 (2017-2021). For rows where data is available, skill scores for the time period 2002-2021 are shown in parentheses. The first half (a) refers to predictions for the number of damage claims where $POD_M$, $FAR_M$, $FH_M$ refer to events with $>100$ building damage claims (for car threshold refer to text) and the second half (b) to predictions of the damage sums in CHF where $POD_M$, $FAR_M$, $FH_M$ refer to events with $>100'000$ CHF building damages (for car threshold refer to text). The column "No. of events" refers to the number of events that exceed the damage threshold and are considered for the skill metrics.

| Exposure | Hazard | $FAR_M$ | $POD_M$ | $FH_M$ | No. of events |
|---|---|---|---|---|---|
| (a) No. of damage claims | | | | | |
| buildings | $E_{kin}$ | 33% | 87% | 64% | 112 |
| buildings | VIL | 43% | 94% | 58% | 125 |
| buildings | dBZ | 36% | 95% | 63% | 120 |
| buildings | MESHS | 9% (27%) | 94% (94%) | 91% (73%) | 78 (212) |
| buildings (scaled) | MESHS | 11% (30%) | 96% (95%) | 90% (71%) | 81 (232) |
| cars | MESHS | 7% | 74% | 74% | 121 |
| (b) Damages | | | | | |
| buildings | $E_{kin}$ | 65% | 78% | 34% | 284 |
| buildings | VIL | 57% | 78% | 41% | 229 |
| buildings | dBZ | 65% | 71% | 34% | 241 |
| buildings | MESHS | 14% (37%) | 80% (74%) | 77% (60%) | 116 (320) |
| buildings (scaled) | MESHS | 18% (39%) | 81% (75%) | 75% (58%) | 122 (331) |
| cars | MESHS | 8% | 60% | 60% | 233 |

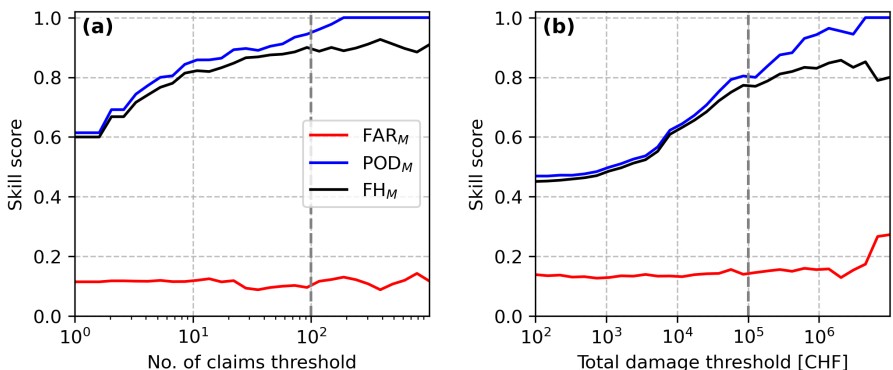

**Figure 9.** Dependence of the skill metrics $POD_M$, $FAR_M$, $FH_M$ to the chosen threshold for the minimum **(a)** number of claims for buildings and **(b)** minimum building damage per event. The threshold used for the calculations in Table 3 is marked with a grey dotted line.

## 5.4 Limitations of the MESHS-based model and potential of crowdsourced data

As this section addresses limitations regarding the spatial structure of MESHS-based hail damage estimates, we focus only on building damage data where exact spatial coordinates are available. Due to the high POD of MESHS, most (88%; see Sect. 3) observed hail damages are captured by the model. The model represents the overall spatial structure of hail damages well for both multiple isolated hail cells and large hail streaks from one thunderstorm (Fig. 10). However, compared to modelled damages, spatial patterns of actual hail damages tend to be narrower and locally more intense mostly contained in 3–10km-wide hail streaks (Fig.A1). Thus, there are areas with MESHS signal but no reported damages despite exposed buildings (e.g. southwest Berne on 26 June 2020, or northern Zurich on 28 June 2021; Fig. 10). The mentioned examples, as well as the FAR above 50% even for high MESHS (Fig. 4), underline the limitations of currently available radar-based hail observations. In particular, they emphasize the known overprediction of MESHS (Schroeer et al., 2023) which is expected from the theoretical limitations of indirect hail size estimations from single-polarization radars (Sect. 1).

For particularly severe events, such as 28 June 2021 (see also Kopp et al., 2022), local extreme damages from roof-penetrating hail are not well represented by the MESHS-based model (Fig. 10). In fact, the underestimated extreme damages are partly compensated by a larger spatial extent, leading to a total damage estimate with the correct order of magnitude. Due to the differing distribution of actual and modelled hail damages within one storm footprint, model skill decreases for smaller scales (e.g. Canton) which may not contain the whole storm footprint. While Canton- or gridcell-level damage estimates could be improved with more accurate hazard data, per-building damage estimates remain out-of-scope when using one impact function for all building types. Nevertheless, model outputs may be used as input for a statistical model to estimate damages on a building level (Miralles et al., 2023).

Our analysis based on long-term and extensive damage data suggests that the spatial extent of damaging hail streaks on the ground are actually narrower than the radar-derived MESHS footprints (Fig. A1). Consequently, long-term averages of MESHS-based damage estimates are spatially smoother and depend more on the distribution of exposure values (Fig. 2) than long-term averages of actual damages (Fig. 11). While some densely populated areas (e.g. the cities of Biel, Thun, Lucerne and the greater Zurich area) are visible as spots with high total hail damages in both modelled and observed long-term damage patterns, others (e.g. Berne and Winterthur) are only clearly visible in modelled damages. This indicates that multiple hailstorms may have either passed nearby with MESHS extending over the city, while the actual hail streaks missed densely populated areas, or thunderstorms with high MESHS but no damaging hail were observed over the city. The severe hail streak on 28 June 2021 in the canton of Lucerne is evident in the observed 20-year damage record (Fig. 11a). It was characterized by extreme total hail damages in a rural area where only relatively few buildings are located (Fig. 2a). Since MESHS-based damage estimates suffer from potentially over-estimating the spatial extent of a hail streak and consequently have lower per-area damage estimates, individual extreme events such 28 June 2021, are not directly visible in the long-term modelled hail damages (Fig. 11b).

While from a nowcasting-perspective, a spatial over-estimation of the potential hail hazard is not necessarily disadvantageous or may even be intended for the purpose of, e.g., warnings, this is not practical for damage modelling. Having learned about the

shortcomings of using radar-based hail intensity estimates, such as MESHS, as a hazard variable for damage modelling, it is worth exploring improvement options. One possibility to improve the hail hazard variable is the usage of crowdsourced reports (Appendix B; Barras et al., 2019). An exploratory hail impact estimate based on crowdsourced data of the extreme hail event on 28 June 2021 in Fig. B2 shows a more spatially accurate hail damage footprint with more realistic extreme local damages, compared to radar-based estimates (Fig.10. This improvement is caused by narrower footprints and fewer occurrences of high hazard intensities, allowing the crowdsource-based impact function for PAA to reach higher values of up to 40% (not shown), compared to 10% for MESHS. Thus, crowdsource-based hail damage estimates have a reduced low-intensity large-area bias. However, drawbacks include inconsistent reporting behaviour (particularly "joke" reports of very large hail), lower accuracy at night, and short data records (2017-2021). Current pre-processing still cannot filter out all false reports of very large hail (Barras et al., 2019), which is crucial for accurate damage modelling. Due to these false alarms and numerous missed events (e.g. nocturnal hailstorms), the model skill of crowdsource-based hail impact estimates over all events (not shown) remains much lower than for radar-based impact estimates. Nevertheless, there is large potential for hail damage modelling based on crowdsourced data, especially because areas with high exposure (buildings or cars) mostly also have high population density and many crowdsourced reports. To fully exploit this potential, dedicated pre-processing of hail damage reports needs to be developed, which is a promising direction for future research.

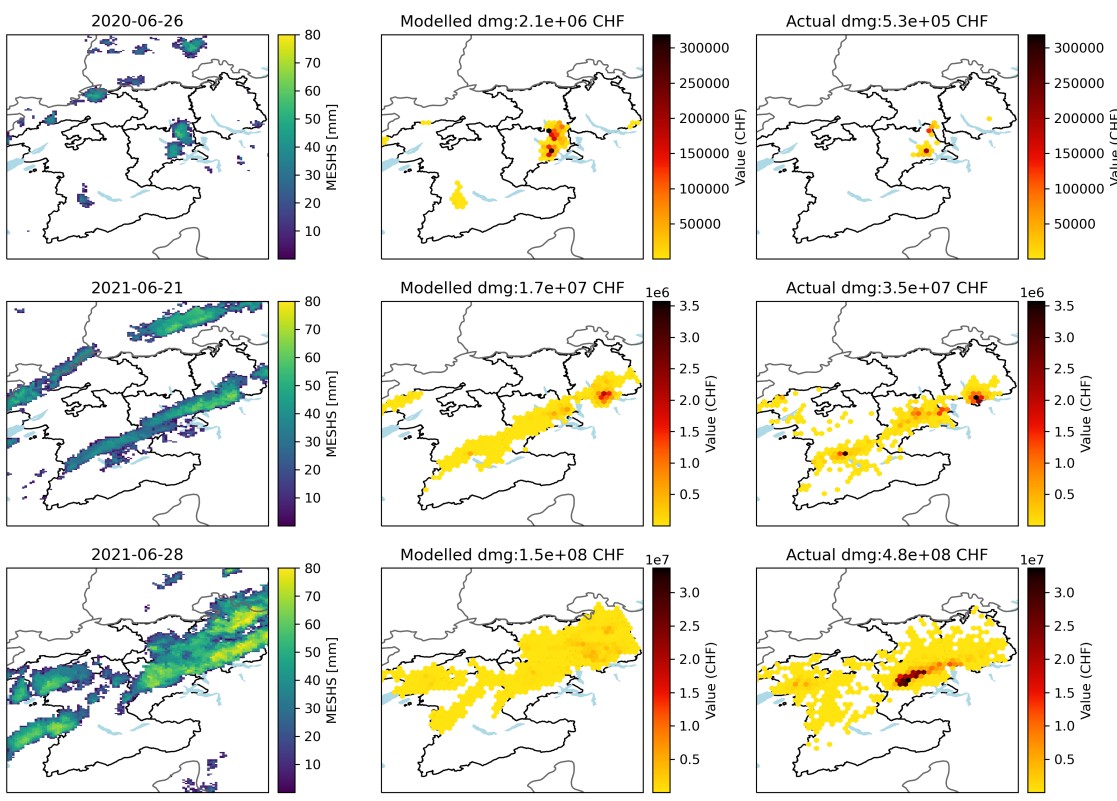

**Figure 10.** Example of model performance for three hail events on 26 June 2020, 21 June 2021, and 28 June 2021. Shown are (left) MESHS, (middle) modelled, and (right) reported building damages. Note the different damage value scales for each event.

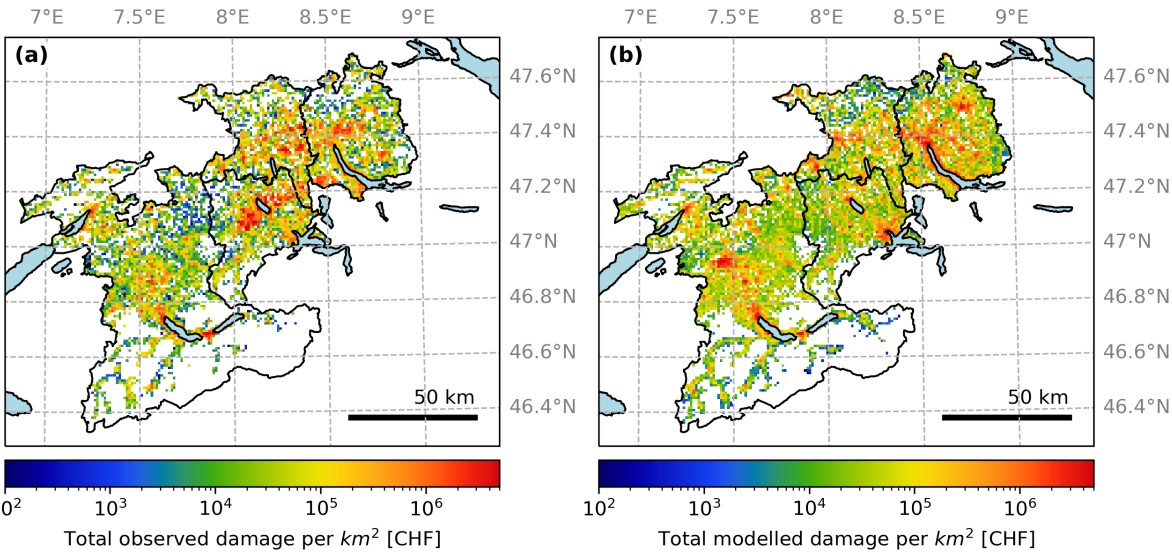

**Figure 11.** Long-term observed and modelled hail damages to buildings: **(a)** Total damage over 20 years per $1\,km^2$ gridcell. **(b)** Total modelled hail damage over 20 years, based on MESHS data.

## 6    Discussion

**Radar verification with insurance claims.** The available geolocated hail damage data, based on building insurance claims, reveal coherent hail streaks with damaging hail where a maximum hail size of >2.5cm can be expected based on known vulnerabilities of building materials (Stucki and Egli, 2007). In combination with complete exposure data, the insurance claims constitute a particularly useful verification metric in contrast to individual reports (e.g. ESWD; Púčik et al., 2017), which do not allow a quantification of false alarms (Delobbe and Holleman, 2006). While many studies have used hail damage data to verify radar-based hail detection (Hohl et al., 2002b; Holleman et al., 2000; Kunz and Puskeiler, 2010; Skripniková and Řezáčová, 2014; Nisi et al., 2016), only few have used building-scale data (Schuster et al., 2006; Kunz and Kugel, 2015; Brown et al., 2015; Warren et al., 2020) and to our knowledge, with 250'000 claims, we base our work on the largest number of per-building hail damage reports. A limitation of insurance claim data is their inaccurate temporal reporting (e.g. Warren et al., 2020), which requires a plausibility check (Sect. 2.2.1). Thus, the data is not fully independent of radar-based hazard variables, but the rare occurrence and small spatial extent of hail streaks allow for a pre-processing approach with high confidence, as there are only few instances where hail damages in a given location are plausible on multiple days near the reported date (Sect. 2.2.1).

**Reflection on the model calibration approach.** Compared to earlier radar-based hail damage modelling in Switzerland (Hohl et al., 2002a, b), we calibrate our model with spatio-temporally complete data rather than individual storms. Using all data leads to more robust results and direct usability with operational radar data as input. However, we acknowledge that radar signal attenuation can lead to a bias in individual hail cells, but we expect this bias to be negligible compared to the inherent uncertainty of (large) hail detection from radar data (e.g. Blair et al., 2011).

The chosen spatially explicit calibration approach reveals the underlying long-tailed distribution of actual hail damages (Fig. 5), given a certain hazard intensity. Sampling from this distribution instead of directly using the mean PAA or MDR to model hail damages was considered, but finally not implemented. As visible in the shown examples (Fig. 4d–f, 10, and A1), the extreme PAA and MDR values are typically concentrated in one hail streak which is a subset of the MESHS footprint. Its location within the MESHS footprint often does not correspond to the highest hazard intensity derived from radar data. Thus, while a random sampling may improve per-gridcell statistics, the modelled hail damage footprints would not be improved towards the observed hail streaks, and aggregated damages would remain unaffected.

**Horizontal hail drift.** Some studies explicitly consider horizontal drift and shift radar footprints to obtain an optimal overlap to insurance claims, which improves correlations (Hohl et al., 2002a; Schuster et al., 2006). Here, we consider this hail drift in the insurance claims pre-processing, and by analyzing hail occurrence skill scores with a correlation-maximizing unidirectional shift of hazard variables (Sect. 3). For the model calibration and evaluation, no spatial shift is considered for two main reasons. First, the explicit hail drift only marginally improves skill scores in Fig. 4 which suggests that MESHS generally spatially overpredicts hail streaks on the ground, rather than suffering from wind-related directed shifts. Secondly, most insurance claims (88%) are within a MESHS footprint without considering a spatial shift. Furthermore, an optimal shift (as in Hohl et al., 2002a; Schuster et al., 2006) could not be applied in a real-time application, before damage claim data is available.

**Model evaluation and applicability.** The developed model focuses on representing aggregated hail damages to buildings and cars, based on the exposed number of assets and their value. While different types of used building materials, building age, and spacing between buildings strongly influence the vulnerability of an individual building (Hohl et al., 2002a; Stucki and Egli, 2007; Schmidberger, 2018), the developed impact function represents a mean vulnerability which yields realistic estimates on an aggregated spatial scale. Exploratory results show that e.g. newer buildings tend to be more vulnerable with an average PAA of 9% at MESHS of 60 mm for buildings built after 2002 vs. 5% for all buildings (Fig. 6). However, the per-event model skill remains largely unaffected when using separate impact functions for different classes of construction year. Similarly, the consideration of a scaled building value that approximates the value of exposed building parts does not improve model skill (Table 3).

While the imprecise nature of available radar metrics with the high FAR (Sect. 3) makes a direct quantification of the influence building attributes challenging, the lack of model skill improvement when considering more details of the exposure data suggests that currently the overall large uncertainty in the hail damage model mainly stems from the available radar metrics. Since none of the considered single-polarization radar variables accurately and consistently distinguishes large hail from smaller (not damaging) hail, damage estimates have high uncertainty. While uncertainties of over one order of magnitude are common in natural hazard risk modelling due to the complexity of modelled processes (e.g. Röösli et al., 2021; Eberenz et al., 2021; Lüthi et al., 2021), it has important implications for the model applicability.

The presented MESHS-based hail damage model is suitable to provide approximate estimates for the number of claims and total cost of building and car damages for both a user-provided portfolio, or an exposure layer including all asset values, e.g. approximated through nightlight intensities (LitPop; Eberenz et al., 2020). Note that uncertainties increase with a smaller spatial extent of the exposure portfolio (Sect. 5.4). Assuming the ratio between building/car values and repair costs is similar between countries, the impact functions derived in this study can be used for other regions. A scaling of the impact function to reflect different building materials could be explored, but our results suggest that the uncertainty in MESHS exceeds the differences in exposure properties, such as building volume and year of construction, when analyzing aggregated hail damage estimates. The areas with expected hail damages are well covered, but extreme local impacts are not well represented, since modelled hail damages present with larger spatial extent and lower local impact. Thus, for applications where an accurate representation of intense hail streaks is crucial, statistical post-processing (Miralles et al., 2023) or a more accurate hazard layer is indispensable. A data product for the latter is not yet available in Switzerland, but promising work is currently underway to derive more accurate hail measures, e.g., from a combination of radar and crowdsourced data or from improved hail detection algorithms based on dual-polarization radar signatures.

## 7  Conclusions

This study quantifies observed hail damages to buildings and cars in Switzerland, utilizing the damage data to verify existing radar-based hail intensity measures and to build an open-source model which predicts hail damages based on radar data. Severe hail causes large building damages in the selected four cantons of Switzerland with 1.39 billion CHF in the last 20 years, averaging 1'400 CHF per building and including 1187 claims above 100'000 CHF. 90% (1.25 billion CHF) of the total damage to buildings occurred during the strongest 30 hail events, with a single event on 28 June 2021 causing 35% (483 million CHF). Hail damages are mostly concentrated in 3–10 km wide hail streaks with consistently reported damage claims in all gridcells with exposed buildings.

Of the investigated radar-based hail intensity measures, the maximum expected severe hail size (MESHS; Betschart and Hering, 2012) performs best, with a high probability of detection of 60% on a gridcell-level and 88% for individual damage claims. However, a substantial false alarm ratio of over 50% even for extreme MESHS values indicates that high MESHS often occur without damaging hail, confirming that MESHS should not directly be interpreted as local hailstone size, but as "maximum expected" size, given the storm intensity.

Using a novel empirical calibration approach, which informs about the shape of the impact function and provides detailed uncertainty estimates, we calibrate impact functions for buildings and cars depending on the available single-polarization radar metrics: MESHS, reflectivity, hail kinetic energy, and vertically integrated liquid. Spatially explicit hail damages are estimated for each day from April–September 2013–2021 by combining the impact function with exposure and hazard data. The MESHS-based model successfully estimates the correct order of magnitude for most daily hail damages across four cantons in Switzerland, which allows deriving first-guess estimates of expected damages that can serve as valuable information for, e.g., insurances in taking immediate action after large hail events. In particular, the largest observed 26 car damage events and the largest 33 building damage events are modelled with the correct order of magnitude. However, considerable uncertainty remains, especially in the spatial structure. The model has a bias towards large extent and lower local impact of hail damages, reflecting the fact that MESHS (or any other available radar variable) does not reliably distinguish medium-sized hail from extreme hail which causes roof-penetrating hail damages. The uncertainty in hail damage predictions with the proposed radar-based model remain large, warranting careful assessment of its intended use. While the model is suitable to derive approximate hail damage estimates to buildings and cars in near real-time, accurately quantifying tail risk using a probabilistic event set of MESHS footprints remains challenging given limitations in the radar-based identification of extreme hail.

Crowdsourced data shows promising results regarding the modelling of locally intense hail damages, but longer time series and improved pre-processing are required to outperform radar-based estimates. Furthermore, the development of dual-polarization hail intensity measures could potentially overcome some limitations regarding the spatial structure of intense hail streaks. Such innovations will have instantaneous effect, as new variables can easily be included in our hail damage model using the presented calibration approach. Lastly, the use of hail size estimates from numerical weather predictions as hazard variable would allow for new applications of the model. For example, ensemble weather predictions may be used to provide proba-

bilistic hail impact estimates as a basis for impact-based warnings. Furthermore, application of the model to high-resolution

convection resolving climate simulations could quantify changing hail risk in a warming climate.

*Code and data availability.* CLIMADA is an open-source and -access software (https://doi.org/10.5281/zenodo.7691855) and can be used with any user-provided portfolio under the General Public Licence gpl-3.0. The code for the conducted analysis and creation of all figures is written in Python 3.9 and can be found in the CLIMADA paper repository under the following link: https://github.com/CLIMADA-project/ climada_papers. Radar data is available from MeteoSwiss upon request (https://www.meteoschweiz.admin.ch/service-und-publikationen/

service.html), with a licencing requirement for commercial use. Hail damage data for both buildings and cars from insurance companies are only available within the scClim project (https://scclim.ethz.ch/).

## Appendix A: Building damages

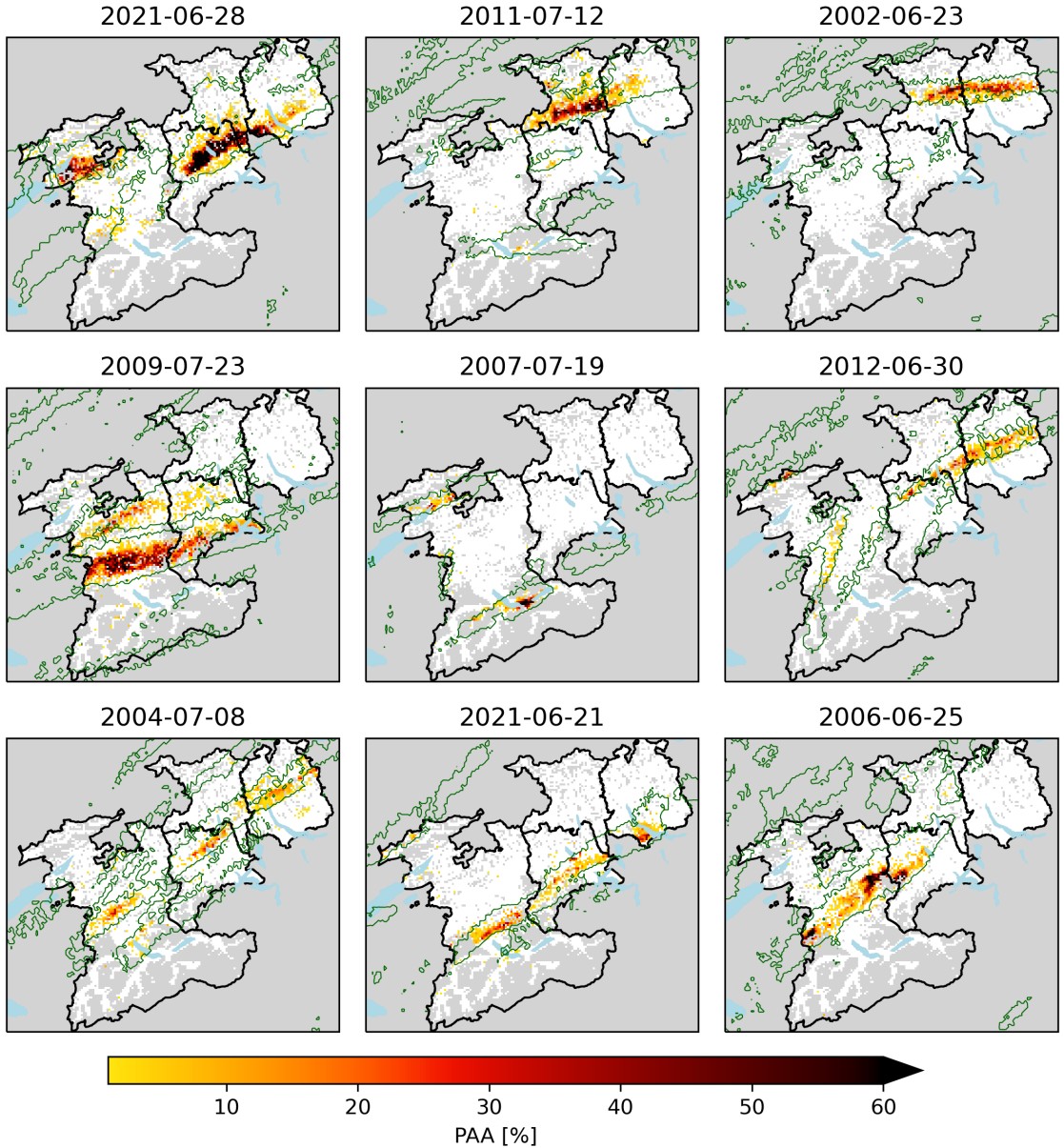

**Figure A1.** Percentage of damaged buildings for the 9 strongest hail events between 2002 and 2021. 1 km gridcells with less than 10 buildings are marked in grey and green lines outline the 20 mm MESHS contour.

## Appendix B: Crowdsourced data

MeteoSwiss launched a crowdsourcing function in 2015 via the MeteoSwiss app with 500'000 active daily users. Since 2017, users can select the following hail size categories in their report (Barras et al., 2019):

- Smaller than coffee bean: >5 mm

- Coffee bean: 8 mm (5-15 mm)

- One Swiss franc coin: 23 mm (15-27 mm)

- Five Swiss francs coin: 32 mm (27-37 mm)

- Golf ball: 43 mm (37-55 mm)

- Tennis ball: 68 mm (>55 mm)

Multiple plausibility filters are applied to remove implausible reports. Firstly, reports outside a conservative threshold of 35 dBZ radar reflectivity within a buffer zone of 4 km are removed. Furthermore, entries from user ID's with an unusual reporting pattern are removed (Barras et al., 2019).

From the data we created a gridded dataset, with the following approach: For each 1 km grid point, the average size and total number of crowdsourcing reports are computed. Gridcells with a population of >2000 people and no crowdsourcing reports are assumed to have no hail (0 mm) because observations show that within known hail footprints, gridcells with >2000 people consistently contain hail reports. For grid points with a population of less than 2000 people and no hail report, the hail size is estimated by considering the average of all cells within 3.5 km distance, given that this area contains at least three reports. Lastly, a noise filter is applied which estimates the hail size at every gridcell as an average over the surrounding 3×3 km box (i.e. 9 gridcells in total). As the largest hail size report is "Tennis ball" with 68 mm, higher values cannot be reached.

Figure B1 shows all hail reports of 28 June 2021 filtered as described in Barras et al. (2019). A comparison with areas of high reported hail damages shows that only within areas of high PAA "Tennis ball" hail stones are consistently reported, although scattered "Tennis ball" reports in other regions also exist. These may be individual large hail stones that did not hit any building or, more likely, reports where a user overestimated the hail stone size. Note that the category "Tennis ball" was originally introduced to filter our "joke" reports of users who choose the largest possible category for fun (Barras et al., 2019). Applying the calibration approach described in Sect. 4 to the gridded crowdsource dataset, corresponding empirical impact functions are derived. Figure B2 shows the resulting modelled hail damages for 28 June 2021, which represent the observed locally intense hail damages well.

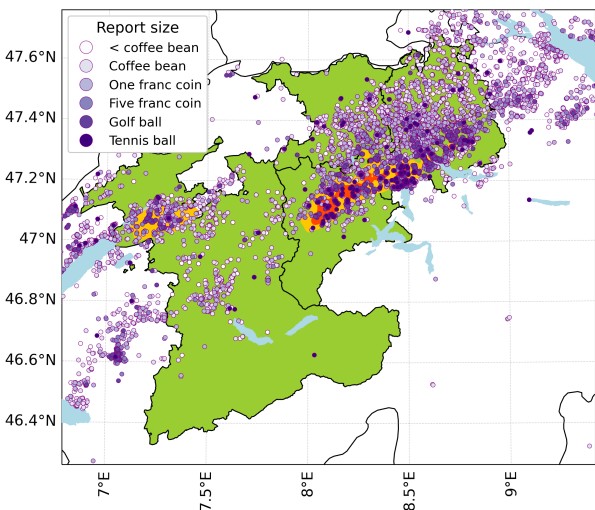

**Figure B1.** Filtered crowdsourced reports from the extreme hail event on 28 June 2021, with shading marking areas of PAA>10%(orange) and >50% (red). Cantons with available building damage data are marked in green.

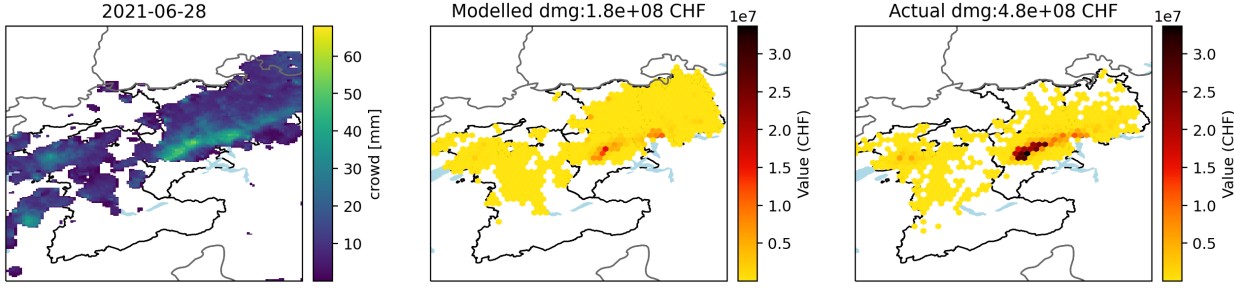

**Figure B2.** Crowdsource-based hail size estimate derived with the gridding approach described in Appendix B and corresponding modelled and observed damages for 28 June 2021. Modelled damages are calculated with an impact function derived as shown in Sect. 4 and crowd-sourced data as hazard variable.

**Appendix C: Model evaluation**

**Table C1.** Skill scores for calculating hail damages to buildings (cars) with different variable combinations for the period 2013-2021 (2017-2021). All values are calculated with the sigmoidal impact function and, as comparison, values calculated with a smooth-empirical fit are shown in parentheses. The first half (a) refers to predictions for the number of damage claims where $POD_M$, $FAR_M$, $FH_M$ refer to events with $>100$ building damage claims (for car threshold refer to text) and the second half (b) to predictions of the damage sums in CHF where $POD_M$, $FAR_M$, $FH_M$ refer to events with $>100'000$ CHF building damages (for car threshold refer to text). The column "No. of events" refers to the number of events that exceed the damage threshold and are considered for the skill metrics.

| Exposure | Hazard | $FAR_M$ | $POD_M$ | $FH_M$ | No. of events |
|---|---|---|---|---|---|
| (a) No. of damage claims | | | | | |
| buildings | $E_{kin}$ | 33% (53%) | 87% (89%) | 64% (46%) | 112 (163) |
| buildings | VIL | 43% (43%) | 94% (94%) | 58% (58%) | 125 (125) |
| buildings | dBZ | 36% (43%) | 95% (96%) | 63% (57%) | 120 (134) |
| buildings | MESHS | 9% (9%) | 94% (96%) | 91% (91%) | 78 (80) |
| buildings (scaled) | MESHS | 11% (11%) | 96% (96%) | 90% (90%) | 81 (80) |
| cars | MESHS | 7% (7%) | 74% (77%) | 74% (76%) | 121 (122) |
| (b) Damages | | | | | |
| buildings | $E_{kin}$ | 65% (65%) | 78% (74%) | 34% (34%) | 284 (263) |
| buildings | VIL | 57% (61%) | 78% (81%) | 41% (39%) | 229 (261) |
| buildings | dBZ | 65% (67%) | 71% (70%) | 34% (32%) | 241 (239) |
| buildings | MESHS | 14% (21%) | 80% (88%) | 77% (77%) | 116 (132) |
| buildings (scaled) | MESHS | 18% (24%) | 81% (88%) | 75% (74%) | 122 (138) |
| cars | MESHS | 8% (8%) | 60% (66%) | 60% (66%) | 233 (237) |

## Appendix D: Cross Validation

To evaluate the model performance on independent verification data, a 5-fold cross validation (CV) is performed on the building damage model. Impact functions are calibrated 5 times, with only 80% of the building damage data, and verification statistics are calculated on the remaining 20%, which correspond to a different 4-year chunk for each CV split (Table D1). The resulting impact functions (Fig. D1a,b) are all within the bootstrap CI and remain close to the best fit, with the MDR impact function of CV split 5 being an outlier. The MDR of this split is over 50% lower than the best fit (Fig. D1b) because it does not include the extreme hail damages of 2021 in its calibration data. However, despite the substantially lower impact function, skill scores in Table D1 are actually even higher compared to the best-fit impact function, which is calibrated on the full dataset. While the MDR impact function of CV split 5 underestimates the most severe events, some previously overestimated events are better captured with the lower impact function. In addition, damage estimates of a few small impact events increase, as they are dominated by MESHS values below 35 mm, where the impact function of CV split 5 is marginally higher than the default. With the slight increase in modelled damages, four previously 'missed events' are within an order of magnitude, increasing the $POD_M$ by 6%. Similarly, the modelled number of damaged buildings in CV split 1 decreases for some events dominated by MESHS values between 20 and 40 mm, where the impact function is lower (Fig. D1a), decreasing the $FAR_M$ by 6%.

For all other CV splits, skill scores never vary more than 5% between the best-fit impact function and the corresponding CV impact function, which is independent of the validation data. Thus, it is concluded that the model performance is comparable on independent verification data. In contrast, the absolute performance on different 4-year validation splits differs strongly. For the first two CV splits (until 2009) model skill is lower than for the whole dataset, CV split 3 (2010-2013) is comparable, and CV split 4&5 (2014-21) show improved performance. This result further highlights the discussed improvement of model predictions with the newest (dual-polarization) radar generation, which were installed in 2012 during CV split 3 (Sect. 2.1 and 5).

A corresponding CV analysis with car data was performed, with each CV split containing one out of the five years of available data (2017-2021). Results (not shown) yield analogous conclusions as with building data: (1) CV impact functions are captured within the bootstrap CI, (2) all skill scores deviate less than 5% from calculations with the best fit impact function, and (3) the impact function with 2021 excluded from training data is at the lower bound of the bootstrap CI.

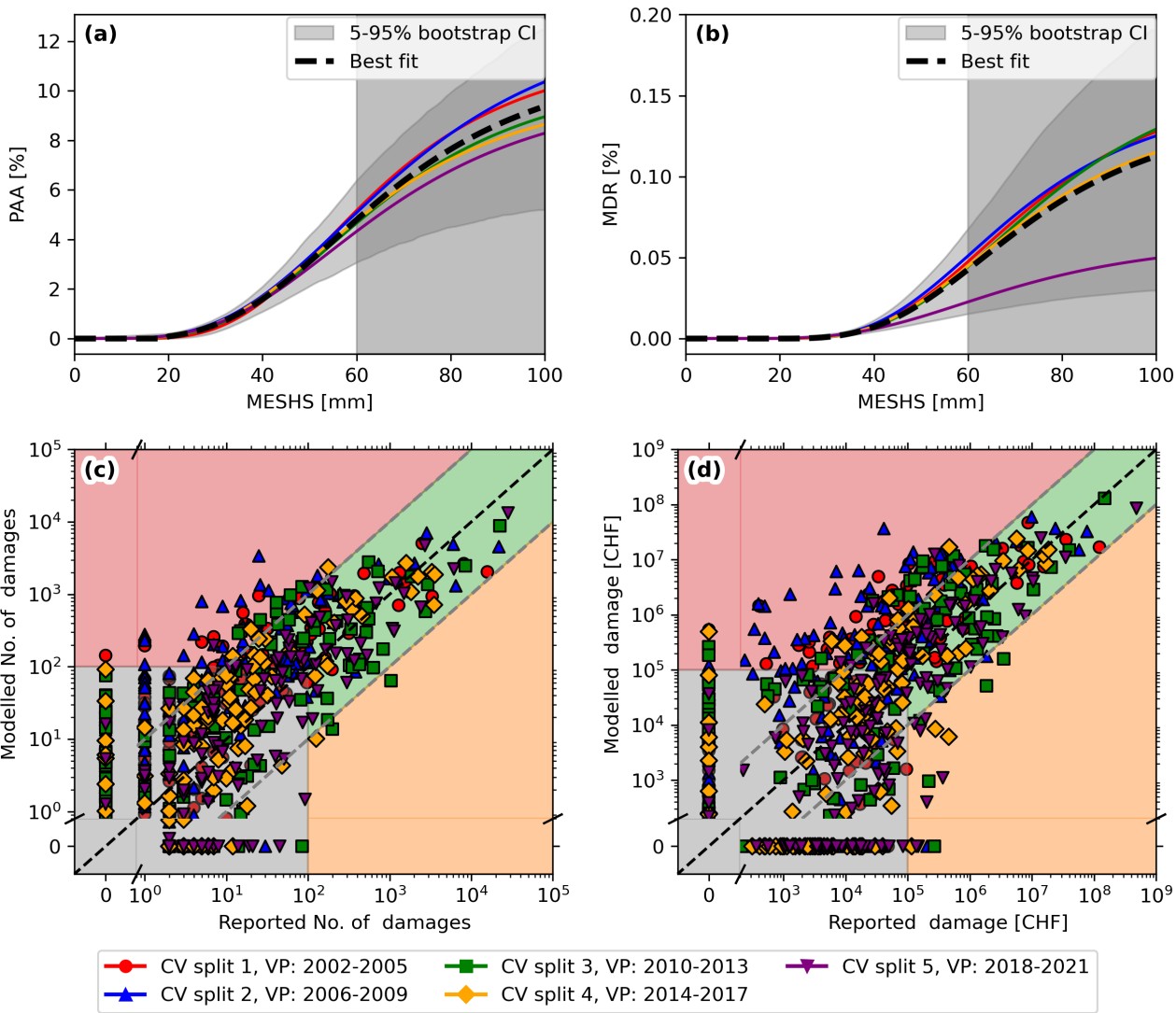

**Figure D1. (a,b)** Sigmoid-type impact functions calibrated for 5 cross validation (CV) splits. Each one is calculated using calibration data from all years except the corresponding validation period (VP). In black, the best fit (calibrated with all data) is shown and the 90% bootstrap confidence interval is shaded grey. **(c,d)** Modelled vs. reported damages as in Fig. 8, where modelled damages for each VP are calculated with the corresponding impact function in (a,b), which is calibrated excluding data from the VP.

**Table D1.** Skill scores as shown in Table 3, but per cross validation split. Values in parentheses are skill scores calculated on the same 4 years of verification data, but using the impact function calibrated on the whole dataset ('best fit' in Fig. D1a,b).

| CV split | $FAR_M$ | $POD_M$ | $FH_M$ | No. of events | VP |
|---|---|---|---|---|---|
| (a) No. of damage claims | | | | | |
| 1 | 30% (36%) | 100% (100%) | 71% (65%) | 41 | 2002-2005 |
| 2 | 49% (49%) | 91% (91%) | 52% (53%) | 61 | 2006-2009 |
| 3 | 18% (18%) | 93% (93%) | 80% (80%) | 45 | 2010-2013 |
| 4 | 11% (12%) | 89% (89%) | 86% (86%) | 29 | 2014-2017 |
| 5 | 6% (6%) | 96% (96%) | 95% (95%) | 38 | 2018-2021 |
| (b) Damages | | | | | |
| 1 | 45% (45%) | 79% (79%) | 54% (54%) | 67 | 2002-2005 |
| 2 | 60% (56%) | 59% (61%) | 39% (43%) | 95 | 2006-2009 |
| 3 | 30% (30%) | 73% (73%) | 65% (65%) | 60 | 2010-2013 |
| 4 | 20% (21%) | 81% (81%) | 74% (74%) | 43 | 2014-2017 |
| 5 | 10% (11%) | 82% (76%) | 81% (75%) | 62 | 2018-2021 |

*Author contributions.* **TS:** Conceptualization, data curation, methodology, software, visualization, writing - original draft, writing - review & editing. **RP:** Methodology, software, writing - review & editing. **LV:** Supervision, writing - review & editing, **KS:** Supervision, writing - review & editing, **DB:** Conceptualization, funding acquisition, supervision, writing - review & editing

*Competing interests.* The authors declare no competing interests.

*Acknowledgements.* We thank Alessandro Hering, Cornelia Schwierz, and Urs Germann from MeteoSwiss for useful inputs regarding the meteorological data. Further, we are greatful to the scClim project partners from the insurance industry for the provided data and explanations thereof. Specifically, we thank Mirco Heidemann, Jan Wüthrich, and Daniel Steinfeld from GVZ, Markus Wigger from GVL, Daniel Wey and Manuel Vonarb from AGV, and Hannes Suter from GVB. Finally, we thank all scClim (https://scclim.ethz.ch/) researchers and Jérôme Kopp for valuable inputs throughout the project. This study was funded by the Swiss National Science Foundation (SNSF) Sinergia grant
CRSII5_201792.

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
