# Peer review of "An open-source radar-based hail damage model for buildings and cars"

_Natural Hazards and Earth System Sciences, 2023_

## Referee Comment (RC1)

All in all, a nice paper on the development of a hail risk model (open source) for buildings and cars in Switzerland; interesting to read, well structured, definitely an added value for the scientific community; even if I did not fully understand the development of the risk model. However, I have one major fundamental critical point (see major comment 1) that needs to be clarified before final publication.

**Major comments:**

1. **The main critical point of the paper**: If I understand it correctly, the same data are used for the development AND for the validation. So it is logical that the result should be "halfway" good; usually it is common to have a data set with which a model development is done (e.g. 80% of the data set) and with the remaining 20% (or x years) a validation is done. This needs to be properly clarified!

2. Section 5: This is an important but "hard to digest" section - request to the authors to work again on the structure of the section and to take the reader by the hand better; it jumps a bit back and forth between the figures/tables without taking the reader fully with each figure (figure describes what is shown, but without going into the "results" (interpretation)); also, Figure 8 could be discussed in more depth.

3. Why is a model development with the other "hazard" parameters still being done; what is the "added value"?

4. The general reader (or non-European) cannot understand a CHF; i.e. it is unclear how much a CHF is worth. Suggestion to introduce conversion into EURO and/or US dollar; or maybe convert all CHF values into EURO.

**Minor comments:**
See also some small/further comments/suggestions direct in the pdf-Version.

Introduction: Paper-Suggestion: Ackermann et al. (20239: doi:10.5194/amt-2023-161

L75/76: It is unclear what CLIMADA is; needs to be explained (or introduced).

L79: You have also a section 6 and 7 → one sentence

L154-162: Are there any experiences from the literature with similar results / discussions?

L191: Why 10? Did you do sensitivity studies?

L196: Why HSS? Is it good :) Can it be explained by literature? Not everyone familiar with this metric...

Figure 4: Suggestion: Introduce a, b, c, d, e, f; this also makes it easier to reference in the text. Lower panale: Word "examples" missing

L241: Would it be worthwhile to develop a damage model based on all three parameters (combination)?

L241: Why is it even worth discussing/including the other two parameters in the publication if the other two are not so "good"?

L283ff: What is the reason for this? Resistance of the buildings?
By the way: Do you notice a difference in the age of buildings in relation to damage? Are new buildings more vulnerable?

L285: „However, the fraction of these false alarms reduces with increasing MESHS while the number of gridcells with high damages of >10% PAA and >0.1% MDR increases"
Is there a better way to describe this result? It took me a long time to understand what was meant.

L307: How exactly do the equations look? Please communicate explicitly that the parameters can be found in the appendix; however, the "basic equation" should be written somewhere.

L315f „the PAA and MDR of the car impact functions include the probability of cars being covered (e.g. in a garage) or located elsewhere"
How do you do that, has that been described?

L347: It would be helpful for the reader if Table 2 is introduced first and only then the link to Table C1 is made (since these are almost congruent).

Table 2 and table C1:
Can these two tables not be combined (since a big part is congruent)?

L390: "limitations"
Reasons? Literature?

L397: "For damages on even smaller scales down to individual buildings, the here developed model is not fit-for-purpose."
Please explain in more detail why not.

L399ff: Is anything similar observed in the literature?

Table D1:
This table is an important result; shouldn't it rather appear in the main part?

[revised manuscript text omitted]

---

## Referee Comment (RC2)

**Review of "An open-source radar-based hail damage model for buildings and cars" by T. Schmid et al.**

**General comments**

Schmid et al. describe the development of a new open-source hail damage model. All aspects are clearly and thoroughly presented: the building of the hail hazard from radar data, its conversion to damages using insurance company claims and exposure, and evaluation of model performance at smaller (1 km$^2$) and aggregate spatial scales. Further, the discussion of model limitations is detailed and objective, providing the reader with some understanding of its weaknesses. The information in this article will be of great benefit to researchers and practitioners in this area.

This reviewer thanks the authors for a well-written article containing lots of new and interesting information, and asks that they consider the points below before finalising their manuscript for publication.

**Major comments**

**1.** The main application of the model is a near real-time estimate of hail damages to help insurance companies respond appropriately to a hail event, and the model evaluation in Section 5 defines success as modelled total claim numbers/damages within one order of magnitude of observed. However, the difference between 5,000 and 500, or 50,000 claims is very significant to the industry, and more precise information would be valuable for potential users. As a suggestion:

- – identify significant events, e.g. those with 500 or more observed or modelled building claims
- – compute the ratio of model-to-observed number of claims for each event
- – report the mean and std dev of this ratio over all events

**2.** The hail hazard from radar is described as the main source of uncertainty in modelled losses (in the Abstract, Conclusions and in sections 4.1 and 6). I think other factors should be considered before concluding on the main source of uncertainty in modelled losses.

Values in Table 1 of Kopp et al. (2022) indicate MESHS resolves the 28 June 2021 event as having a very high rank in history, and they describe MESHS return periods above 70 to 100 years. This suggests radar-based hail does resolve high hazard severity, therefore, the low estimated loss for this severe event must be due to the model's impact functions. This counter-evidence is just one event, however, inspection of the top ten observed events (focussing on the top right corners of plots in Figure 8) shows the model has a low bias for these most severe events, hence the model's impact functions could be responsible for those modelled loss errors which are most material to companies (responding to large hail events). If radar-based hail does resolve high hazard severity, then I can think of two reasons why impact functions fail to convert high hazard to severe loss, as now discussed.

First, the horizontal drift of hail from radar levels to the ground means the calibration of damage *per building* cannot be sharply resolved by radar data. For example, high MESHS will often be collocated

with buildings experiencing smaller hail at the ground, and vice-versa, leading to a much weaker relation between MESHS and damage ratio than actually exists. This possibility may be worth discussion in the text?

A potential second source of weak relation from hazard to damage is the fact that the damage ratio varies with building attributes. The authors explore some possibilities in lines 462 to 470 and find some weak evidence. Have the authors considered splitting buildings by rural and urban settings, and building separate impact functions? Much higher damage ratios in rural regions are a significant feature in the U.S., though I don't know if this applies to Switzerland too.

To be clear, I ask that these possibilities be considered for discussion in the manuscript, while any further model development is at the authors' discretion.

**Minor comments/corrections**

1. line 16: could the damages be expressed in EUR, either as a replacement or as an addition to CHF? A significant fraction of readers may have to break away from the manuscript after the first sentence to check the value of a CHF.

2. In Section 2.2.1, could an annual timeseries of number of claims per year, and mean claim size (both normalised) be included? It may help to understand why model validation in Table 2 is slightly poorer in the first ten years – perhaps higher FAR is due to poorer data collection 10-20 years ago?

3. Figure 4, and text: results using the 4 km buffer could be viewed as more appropriate, whereas those for 1 km cells give the reader an indication of how much hail drift can affect the relation between hail at radar levels and damage on the ground. From this viewpoint, the solid and dashed lines could be swapped in the top row of Figure 4, and the text in Section 3 focuses more on the 4 km buffer data? Irrespective of whether the 4 km buffer is viewed as the standard choice, could the HSS for 4 km buffer data be included in Figure 4?

4. Figures 6 and 7, and text: could the CDR be included in these two figures? The CDR is the damage ratio, conditional on damage occurring, and often referred to as mean severity. CDR is expected to rise with increasing hazard severity but it is not clear this is the case from figures 6a and b, or 7a and b.

5. x-axis legend on figure 8d: typo in 'normalized'

6. line 351: any evidence that the latest radar has fewer small patches? Or is there more complete data collection in the present-day? (related to comment 2 above.)

7. Figure 10: the reader may get more benefit from maps of damage ratio values in the 2nd and 3rd columns, with total loss remaining in the title? The damage ratio reflects MESHS values and might be more interesting than loss values which mainly reflect exposure density.

8. Lines 471-474: results in Table 2 show MESHS clearly distinguishes damaging events (> 100 claims) from non-damaging events. The problem for most users will be that the loss model does not resolve the most severe events in history, and as mentioned above in the second Main Comment, MESHS may resolve high hazard severity but they are not translating to high loss.

---

## Author Comment (AC1)

NHESS-2023-158

**Reply document**

**An open-source radar-based hail damage model for buildings and cars**

*Reply to major revisions of reviewer 1 and 2 in October 2023 by Timo Schmid, Raphael Portmann, Leonie Villiger, Katharina Schröer, and David N. Bresch*

We thank the reviewers for the useful comments and constructive feedback. The suggested changes substantially improve the manuscript, and we addressed all comments in the following document. The comments of the reviewer are shown in black and our replies in blue. We number reviewer comments for referencing purposes throughout the document (comment 1 = C1, etc.). Changes in the manuscript are referenced with the starting line number, where removed parts are crossed-out and *new additions are in italic*. All line numbers refer to the originally submitted manuscript. Figures that only appear in the reply document are labelled as Fig. R1 to R3.

**Reviewer 1**

**Major comments:**

**C1:** The main critical point of the paper: If I understand it correctly, the same data are used for the development AND for the validation. So it is logical that the result should be "halfway" good; usually it is common to have a data set with which a model development is done (e.g. 80% of the data set) and with the remaining 20% (or x years) a validation is done. This needs to be properly clarified!

We thank the reviewer for this comment. Indeed, we used the complete dataset for the model calibration and the verification. As our model only fits 3 impact function parameters it cannot overfit to specific patterns in the data (as e.g. a neural network), but it may still overfit to extreme values of few events. Thus, we provide bootstrapping confidence intervals of the fitted impact functions. However, the bootstrapped impact functions were so far not used in the skill score calculations.

We agree with the reviewer that explicitly showing the model performance on independent verification data further improves the manuscript. To achieve this, we perform a cross validation analysis in addition to the bootstrapped confidence intervals, as it allows for a clear separation of calibration and verification data. Thus, we add an Appendix section where a 5-fold cross validation (CV) is performed. Impact functions are calibrated 5 times, with only 80% of the data, and verification statistics are calculated on the remaining 20%, which correspond to a different 4-year chunk for each cross validation split for the building damage data. The results show that (a) all 5 splits are captured within the bootstrapping uncertainty, and (b) the model performance on the independent verification data is comparable to the skill scores for the full dataset. In fact, for the last 3 of 5 splits the skill scores are even higher with the independent verification data, while for the first 2 splits they are lower. This difference is largely related to the generally observed lower model skill in years before 2012, rather than in differences in the impact function. CV split 5 is an outlier in Fig D1b, as the extreme damages in 2021 are not included in the calibration data. Nevertheless, the model skill in the validation period 2018-2021 remains high at 81% $FH_M$, which exemplifies the small influence of sampling uncertainty on impact functions compared to the overall model uncertainty.

In the paper the Cross validation is referenced as follows:

L280: *Note that Appendix D additionally provides a 5-fold cross validation (CV) analysis which highlights that (a) impact functions for each CV split are well captured with the bootstrapping approach, and (b) that the model skill is equivalent on independent verification data.*

[revised manuscript text omitted]

**C2:** Section 5: This is an important but "hard to digest" section - request to the authors to work again on the structure of the section and to take the reader by the hand better; it jumps a bit back and forth between the figures/tables without taking the reader fully with each figure (figure describes what is shown, but without going into the "results" (interpretation)); also, Figure 8 could be discussed in more depth.

We thank the reviewer for this comment. In order to make section 5 easier to read, we introduce subsections and modify parts of the text. Below is the restructured section 5 with new additions in italics and parts which also relate to other comments marked with brackets [CXX]:

[revised manuscript text omitted]

*Remains unchanged apart from adjustments mentioned in other comments*

**C3:** Why is a model development with the other "hazard" parameters still being done; what is the "added value"?

We thank the reviewer for this question. The additional hazard parameters 'maximum reflectivity' and 'kinetic energy' indeed have lower skill in predicting hail damage occurrence compared to 'MESHS'. As the reviewer points out, this is already evident from section 3 (Evaluation) before the damage model is introduced. However, section 3 focuses exclusively on the occurrence of hail damages and not the intensity; e.g. the percentage of damaged buildings in a gridcell or the relative monetary damage. Thus, it would be theoretically possible, although unlikely, that the additional hazard parameters lead to better estimates of monetary hail damages despite the lower skill on a grid cell level. This is shown not to be the case by additionally calibrating a model with these parameters. However, the question which radar-based product is most suitable for hail damage prediction is often debated, given high overall uncertainties. Consequently, MESHS showing the highest skill among the tested products was not necessarily expected and thus we find that taking this comparative aspect through to the damage model carries valuable information. Furthermore, if a user only has access to e.g. 'maximum reflectivity' data and no 'MESHS' for a given region, they may still use the reflectivity-based impact function despite its lower skill. Note that in many countries with C-band radar MESHS is not operationally calculated.

We adjust the manuscript as follows to highlight these points:

L241: Based on the analysis in the previous section, we here focus on the MESHS-based model. *However, model parameters and skill scores are provided for all hazard variables to accommodate model users without access to MESHS data.*

**C4:** The general reader (or non-European) cannot understand a CHF; i.e. it is unclear how much a CHF is worth. Suggestion to introduce conversion into EURO and/or US dollar; or maybe convert all CHF values into EURO.

We thank the reviewer for pointing this out. We follow the reviewer's suggestion and introduce a conversion to EUR and USD in the reference year 2021. Furthermore, we highlight that since 2008 the CHF is consistently roughly equivalent to the USD (+-15%).

L16: …400 million Swiss francs (CHF) in a single canton (GVL, 2022), with *1 CHF corresponding to 1.09 US dollars and 0.92 EURO in the reference year 2021 (OECD, 2023).*

New Citation: OECD (2023). Exchange Rates, https://doi.org/10.1787/037ed317-en

**Minor comments:**

See also some small/further comments/suggestions direct in the pdf-Version.

**C5:** Introduction: Paper-Suggestion: Ackermann et al. (20239: doi:10.5194/amt-2023-161

We thank the reviewer for highlighting this excellent preprint and mention it in the introduction as follows:

L68:  *Similarly*, Yin et al. (2007) developed an event-based hail risk model for cars in the US, based exclusively on bias-corrected hail reports. Hail losses are estimated by simulating 10'000 years of hail events with a Monte Carlo approach, and using three separate impact function for dent repair labour cost, parts replacement, and depreciation costs. *Lastly, Ackermann et al. (2023) use a deep neural network to successfully estimate relative hail damages to buildings in Australia from S-band radar data and environmental parameters from reanalysis data.*

**C6:** L75/76: It is unclear what CLIMADA is; needs to be explained (or introduced).

We thank the reviewer for this comment and adjust the sentence as follows:

L75:

*Within the open-source risk modelling framework CLIMADA (Aznar-Siguan and Bresch, 2019; Aznar-Siguan et al., 2023), implementing the IPCC (2022) concept of risk, we develop a hail damage model (Sect. 4) with focus on …*

New citation: *Aznar-Siguan et al. (2023) [Code]. https://doi.org/10.5281/zenodo.8308160*

**C7:** L79: You have also a section 6 and 7: one sentence

We thank the reviewer for this comment and add the following sentence:

L79: *Section 6 discusses the results and the model applicability and Sect. 7 summarizes the paper and highlights the key conclusions.*

**C8:** L154-162: Are there any experiences from the literature with similar results / discussions?

We thank the reviewer for this question. From the few publicly available studies which use building-level hail damage reports, the authors are not aware of another analysis of the

statistical relationship between per-building hail damages and building value. Thus, Fig. 3 constitutes a novel result. However, previous studies have identified that the majority of hail damages occur at the building exterior and particularly the roof and blinds (Hohl et al., 2002; Schmiedberger, 2018). Given the damage dependence on the building surface area and the building value dependence on the building volume (correlation of 0.91 in our data), it is to be expected that more expensive and larger buildings report lower relative damages independent of the building material. Using a scaled building exposure, as described in line 161f, is a simple approach to account for this fact. The authors are not aware of a similar approach being used in another publicly available publication, as building-level hail damage data is only available for few studies. However, previous studies (Hohl et al., 2002) have derived hail damage functions for both the damage per building (assuming independence of the building value) and the ratio of building value and hail damage (assuming a 1:1 dependence). When using a scaled building exposure, the dependence of modelled damages on building values is between 'independent' and 1:1 and can be considered a compromise between the two.

**C9:** L191: Why 10? Did you do sensitivity studies?

We thank the reviewer for this question. 10 Buildings were chosen as a suitable trade-off between the number of considered gridcells and the uncertainty within one gridcell. A threshold of 10 buildings retains 75% of all gridcells (see figure below), while increasing the "detection likelihood" of damaging hail 10-fold compared to a cell with one building. Furthermore, below 10 buildings the percent of assets affected (PAA) rapidly increases in uncertainty and has a limited interpretability as e.g. a cell with two buildings can only have PAA values 0, 50, or 100%. While further increasing the threshold reduces the risk of missing damaging hail within one gridcell, the number of considered gridcells decreases rapidly, which increases the uncertainty. The verification was tested with different thresholds between 1 and 100 buildings, with the relative skill of different hazard variables remaining qualitatively unchanged. Absolute values of skill scores consistently increase when only considering gridcells with more buildings, which is expected and e.g. discussed as the 'exposure density'-effect in Portmann et al. (2023). We adjusted the manuscript as follows to explain the threshold selection explicitly:

L191: Each 1km$^2$ gridcell with 10 or more buildings is classified as hail damage (yes/no) on each date. *Different thresholds, ranging from 1 to 100 buildings per gridcell were tested. A threshold of 10 was chosen as it retains 75% of all gridcells, while avoiding conclusions about the hail occurrence based on few buildings only.*

Citation: Portmann et al. (2023).
[Figure]
 https://doi.org/10.5194/egusphere-2023-2598

[Figure]

R1: Fraction of gridcells which exceed a certain minimum number of buildings per cell.

**C10:** L196: Why HSS? Is it good :) Can it be explained by literature? Not everyone familiar with this metric...

We thank the reviewer for this comment and add the following explanation:

L196: ..and Heidke-skill-score (HSS; Wilks, 2019) are calculated as follows:

… and Heidke-skill-score (HSS; Wilks, 2019) are calculated as shown below. The HSS is suitable to quantify forecast skill of rare events like hail storms (Doswell et al., 1990; Wilks 2011) and has previously been used to verify radar-based hail observations (e.g. Kunz and Kugel, 2015; Warren, 2020).

**C11:** Figure 4: Suggestion: Introduce a, b, c, d, e, f; this also makes it easier to reference in the text. Lower panel: Word "examples" missing

We thank the reviewer for this suggestion and adopt the suggested subfigure labelling.

**C12:** L241: Would it be worthwhile to develop a damage model based on all three parameters (combination)?

We thank the reviewer for this question. Indeed, it could be an interesting endeavor to develop a hail damage model that explicitly considers multiple radar-based hail intensity measures, which is currently planned as a MSc thesis project. Initial experiments of combining MESHS and maximum reflectivity into the variable 'weighted MESHS' only led to negligible improvements compared to the MESHS-based model and, thus, are not discussed in the paper. While exploratory analysis shows some complementing information in the three radar variables, they remain strongly correlated which limits the potential for model improvement using all variables. Alternatively (as discussed in Sect. 6 and Sect. 7), we rather suggest investing in the creation of a hazard dataset that incorporates additional, radar-independent data, such as crowdsourced hail reports.

**C13:** L241: Why is it even worth discussing/including the other two parameters in the publication if the other two are not so "good"?

We thank the reviewer for this question. See major comment C3 for the answer.

**C14:** L283ff: What is the reason for this? Resistance of the buildings? By the way: Do you notice a difference in the age of buildings in relation to damage? Are new buildings more vulnerable?

We thank the reviewer for this question. The reason for the large number of gridcells with zero PAA and MDR is mostly related to the limited accuracy of MESHS (see high FAR in Fig. 4). While varying hail resistance of buildings does play a role, spatial patterns of *consistently* observed hail damages within known hail streaks (Fig 4 and A1) indicate that the variability in buildings resistance is a minor influence compared to the high FAR of MESHS. While the limitations of available radar-based proxies dominate the 'false alarm'-gridcells, building vulnerability does play a role regarding the intensity of hail damages. Indeed, there is a difference between vulnerability of older and newer buildings (Discussion section lines 463-467) which is, however, negligible compared to the uncertainty in the hazard data. In order to explain the reason for the large number of gridcells with no damage reports and to rephrase the sentence in C15, we adjust the manuscript as follows:

L232: Spatial footprints in Fig. 4 and A1 show that within an intense hail streak nearly all gridcells consistently contain a damage claim, but no radar variable consistently distinguishes these hail streaks. *The consistently observed damages within a hail streak and the minimal improvement in FAR with explicit hail drift indicate that the false alarms are largely related to*

*limitations in radar-based hail proxies rather than variable building vulnerability or wind-driven hail drift.*

*L283: For all MESHS intervals shown in Fig. 5, many gridcells with buildings exposed to high MESHS values still have zero reported damages (false alarms) and, thus, zero PAA and MDR. The fraction of these gridcells is equivalent to the FAR in Fig. 4 and represents grid cells with buildings where hail is expected according to the radar, but no damage occurred.  Most of these false alarms as well as low damages occur for MESHS below 40 mm (Fig. 5c,d) while high relative damage can be predominantly attributed to MESHS values above 40 mm (e.g. PAA~>20% or MDR~>0.1%; Fig 5c,d).*

**C15:** L285: „However, the fraction of these false alarms reduces with increasing MESHS while the number of gridcells with high damages of >10% PAA and >0.1% MDR increases". Is there a better way to describe this result? It took me a long time to understand what was meant.

We thank the reviewer for pointing this out and rephrased the sentence as shown in comment C14.

**C16:** L307: How exactly do the equations look? Please communicate explicitly that the parameters can be found in the appendix; however, the "basic equation" should be written somewhere.

We thank the reviewer for this comment and add the following reference to the text:

L303: The sigmoidal fit (Fig. 6, in black) agrees well with the empirical PAA and extends it to reach
5-10% at 80mm MESHS. *The sigmoidal function corresponds to Eq. 6 with the parameters from Table D1.*

**C17:** L315f „the PAA and MDR of the car impact functions include the probability of cars being covered (e.g. in a garage) or located elsewhere" How do you do that, has that been described?

We thank the reviewer for this comment. The inclusion of the probability of cars being covered or located elsewhere is a result of directly using the whole car portfolio to calibrate the impact function, rather than an explicit step in the calculation. As we agree, this is not optimally described in the current sentence, we adjust it as follows:

L315:

*As the calibration is conducted with the complete car portfolio, the resulting impact function implicitly considers the probability of cars being sheltered (e.g., in a garage) or spatially displaced. Thus, assuming similar vehicle statistics (e.g., average driving distance or ratio of sheltered cars) as in the calibration dataset, the impact functions can be directly applied to a car portfolio with the location given by the address of each vehicle's most frequent driver.*

**C18:** L347: It would be helpful for the reader if Table 2 is introduced first and only then the link to Table C1 is made (since these are almost congruent).

We thank the reviewer for this comment. See C19 for the implemented adjustment.

**C19:** Table 2 and table C1: Can these two tables not be combined (since a big part is congruent)?

We thank the reviewer for this comment. We acknowledge that a large part of the table is congruent, with Table 2 additionally showing skill scores including the time period before 2012 and Table C1 additionally showing skill scores from a model using smooth-empirical damage functions. After careful consideration, we decided to keep two separate tables to retain a good readability. As suggested in C18 we introduce Table 2 first now:

L347:

Table 2 shows skill scores for hail impact model versions *using the sigmoidal impact function* with different hazard and exposure data. *A comparison to skill scores using the smooth empirical fit (Table C1) shows only minor differences and, thus, further results focus on the sigmoidal impact function.*

**C20:** L390: "limitations" Reasons? Literature?

To give more context, we adjusted the paragraph as follows:

L389:  *The mentioned examples, as well as the FAR above 50% even for high MESHS (Fig. 4), underline the limitations of currently available radar-based hail observations. In particular, they emphasize the known overprediction of MESHS (Schröer et al, 2023) which is expected from the theoretical limitations of indirect hail size estimations from single-polarization radars (Sect. 1).*

**C21:** L397: "For damages on even smaller scales down to individual buildings, the here developed model is not fit-for-purpose." Please explain in more detail why not.

We thank the reviewer for this comment and restructure the whole paragraph to additionally address the reviewers' comments from within the pdf.

L392: *For particularly severe events, such as 28 June 2021 (see also Kopp et al., 2022), local extreme damages from roof-penetrating hail are not well represented by the MESHS-based model (Fig 10). In fact, the underestimated extreme damages are partly compensated by a larger spatial extent, leading to a total damage estimate with the correct order of magnitude. Due to the differing distribution of actual and modelled hail damages within one storm footprint, model skill decreases for smaller scales (e.g. Canton) which may not contain the whole storm footprint. While Canton- or gridcell-level damage estimates could be improved with more accurate hazard data, per-building damage estimates remain out-of-scope when using one impact function for all building types. Nevertheless, model outputs may be used as input for a statistical model to estimate damages on a building level (Miralles et al., 2023).*

**C22:** L399ff: Is anything similar observed in the literature?

We thank the reviewer for this question. While it is known and expected from the design of MESHS that many areas experience smaller hail than the MESHS values (e.g. Schröer et al., 2023; Betschart and Hering, 2012), the visibility of the spatial extent of damaging streaks within a MESHS footprints is a novel result (also see the first paragraph in the discussion). A recent study using the same radar data but focusing on crop damages (Portmann et al., 2023; see reference in C9) show that a MESHS threshold of 20mm results in an overprediction of the extent of damage footprints (for crops) by at least a factor of 2.

**C23:** Table D1: This table is an important result; shouldn't it rather appear in the main part?

We thank the reviewer for this comment and decided to follow the suggestion to move Table D1 into the main part (Section 4).

**Additional changes regarding minor comments of reviewer 1 directly within the pdf**

L15: Switzerland, with a recent extreme event on 28 June 2021 *(see also Kopp et al., 2022)* causing building damages of 400 million Swiss francs (CHF) in a single canton (GVL, 2022).

L28: The local hail intensity *on ground* is primarily expressed…

L63: Furthermore, Schmidberger (2018) developed a hail damage model for *parts of* Germany…

L74: , we conduct a detailed verification of available radar products at 1 km resolution (Sect. 3) .

L84: All hazard data in daily resolution comprise sub-daily information from 6 UTC to 6 UTC *the following day*. The threshold *of 6 UTC* marks…

Fig1: The four cantons (from  *southwest to northeast*: Berne, Lucerne, Aargau, Zurich)

L124: *Furthermore, few addresses (<0.5%)*  *were geocoded with locations outside the cantonal border. Since cantonal building insurances only insure property within their canton these buildings must be incorrectly geocoded and* were removed from the exposure data.

L126: The buildings are distributed over 89% of all 1 km gridcells within the four cantons, *with the remaining 11% containing no building (Fig. 2a). Out of all gridcells with at least one building, 75%* contain ten or more buildings (Fig. 2b).

L129: …of Zurich, Berne, and Lucerne, where land prices are high and   *large multistory buildings are more common* (Fig. 2b).

L133: The total claim volume over the considered 20-year period is 1.39 billion CHF *distributed over 806 days,* of which 90% (1.25 billion CHF) is caused by the strongest 30 hail days

L139: , *Please note that an* accurate temporal reporting of hail damages is not always guaranteed, …

L132: Spatial footprints in Fig. 4 and A1 *exemplary* show that within an intense hail streak nearly all gridcells consistently contain a damage claim

L322: As for the buildings, the simgoidal fit *(Table D1)* is used for subsequent damage modelling.

Fig 7: ~~(a) Impact function for the number of damaged cars, calibrated with data from 2017-2021. The solid green line indicates the 10mm moving average for the empirical percent of assets affected (PAA) and the blue shading its 5-95% bootstrap confidence interval. In black, a sigmoidal fit (Emanuel, 2011) is shown, and pink bars indicate the fraction of damage reports per MESHS intensity. The grey shaded area indicates MESHS values without sufficient data points for a meaningful empirical fit. (b) same as (a), but for damage sums in CHF, with the empirical mean damage ratio (MDR) in blue.~~ *As Fig. 6, but for car damages, calibrated with data from 2017-2021. Due to conditions of the data provider, pink bars show the fraction of damage reports per MESHS intensity, rather than absolute values.*

L385: Due to the high POD of MESHS,  most (88%; cf. Sect. 3) observed hail damages are captured by the model.

L387: However, *compared to modelled damages*, spatial patterns of actual hail damages tend to be narrower and locally more intense , *mostly contained in 3-10km-wide hail streaks (Fig. A1). Thus,*  there are areas with MESHS signal but no reported damages despite exposed buildings …

L412: While from a nowcasting-perspective, a *certain degree of* spatial over-estimation of the potential hail hazard  *is not necessarily disadvantageous or may even be intended for the purpose of, e.g., warnings*, this is not practical for damage modelling.

L429: The available geolocated hail damage data, *based on building insurance claims*, reveal coherent hail streaks with damaging hail where a maximum hail size of >2.5cm can be expected *based on known vulnerabilities of building materials* (Stucki and Egli, 2007).

L437: 2.2.1). Thus, the data is not fully independent of radar-based hazard variables, but the rare occurrence and small spatial extent of hail streaks allow for a pre-processing approach with high confidence, *as there are only few instances where hail damages in a given location are plausible on multiple days near the reported date (Sect.2.2.1).*

L491: *This study quantifies observed hail damages to buildings and cars in Switzerland, utilizing the damage data to verify existing radar-based hail intensity measures and to build an open-source model which predicts hail damages based on radar data.*

L496: *Of the investigated radar-based hail intensity measures,* the maximum expected severe hail size (MESHS; *Betschart and Hering, 2012*)  performs best, …

**Reviewer 2**

**Major comments**

**C24:** The main application of the model is a near real-time estimate of hail damages to help insurance companies respond appropriately to a hail event, and the model evaluation in Section 5 defines success as modelled total claim numbers/damages within one order of magnitude of observed. However, the difference between 5,000 and 500, or 50,000 claims is very significant to the industry, and more precise information would be valuable for potential users. As a suggestion:

– identify significant events, e.g. those with 500 or more observed or modelled building claims
– compute the ratio of model-to-observed number of claims for each event
– report the mean and std dev of this ratio over all events

We thank the reviewer for this comment. Indeed, a damage estimate within one order of magnitude of the actual impacts includes a large spread. Originally, the Root-Mean-Squared-Fraction (RMSF) and a weighted RMSF were included as skill metrics to quantify the ratio between modelled and observed damages. However, the RMSF over all events (without minimum damage threshold) was unintuitive to interpret. While the unweighted version was overly dependent on low-impact events and highly variable, the weighted RMSF was mostly defined by the few largest events and very stable. Thus, RMSF-related skill scores were removed. Nevertheless, we agree with the reviewer that some quantitative information on the model-to-observed ratio may be valuable for model users, particularly for the best-performing MESHS-based model, and conducted a new analysis:

As suggested, we focus on significant events (e.g. L339 and L353ff), but chose the same threshold as for the calculation of the other skill scores (100 buildings, or 100'000 CHF damages) and report the modelled-to-observed ratio for each event. For easier interpretation, ratios below 1 are inverted: i.e. both a modelled loss of half or double the reported damage are quantified with a factor 2. As shown in Fig. R2 (indicated by the dashed lines labelled Q0.5), 50% of the model estimates are less than a factor 2.8 off the reported number of damaged buildings (or factor 4.2 of the reported monetary damages). These results are valid for MESHS-based damage estimates for the period 2013-2021 with the mentioned thresholds. If higher thresholds (e.g. 500 buildings) are chosen the skill further increases and if the time period before 2012 is also included the skill decreases.

In the paper, we report the model-to-observed ratio in the restructured section 5 (as requested in comment C2). We add the relevant paragraphs here without line numbers, as they are part of the restructured section in comment C2:

C2, Sect. 5.1: *For certain practical applications of the damage model, a skill assessment in terms of model-to-observed ratio may be more meaningful than the number of events with modelled damages within one OOM of the observed damages. Thus, for the best-performing hazard variable, MESHS, we additionally quantify how close within the correct OOM the observed damages and model estimates are.*

C2, Sect. 5.2: *In order to quantify how close within the correct OOM hail damage estimates are, we additionally report the model-to-observed ratio. Considering all events after 2012 with more than 100 buildings affected, 50% of the MESHS-based model estimates are less than a factor of 2.8 off the reported number of damaged buildings and 75% less than a factor 5.1. Focusing on all events over 100'000CHF damage, 50% of the MESHS-based monetary hail damage estimates are within a factor of 4.2 off the reported damages and 75% within a factor of 8.7. As with the aforementioned primary model evaluation metrics, the skill in terms of modelled-to-observed ratio decreases for events before 2012 (beige dots in Fig. 8a,b), where more model estimates are further than one order of magnitude off the observed damages.*

**C2, Sect 5.3:** *As expected from Fig 8b,c the model-to-observed ratio tends to be larger for cars than buildings. The 50% quantile is at a factor of 3.8 for the number of damaged cars and 5.4 for the monetary damages, while the 75% quantiles are beyond one OOM. However, when only focusing on the 25 strongest events (containing 80% of all damages) the 50% quantile for both number of claims and monetary damages is below a factor 3 and no events exceeds a factor 8.*

[Figure]

Fig R2: Distribution of the ratio of model-to-observed building damages for the period 2013-2021, where ratios below 1 are inverted. Orange bars refer to the predictions of the number of buildings (PAA impact function) and blue bars to the monetary damages (MDR impact function).

**C25:** The hail hazard from radar is described as the main source of uncertainty in modelled losses (in the Abstract, Conclusions and in sections 4.1 and 6). I think other factors should be considered before concluding on the main source of uncertainty in modelled losses.

Values in Table 1 of Kopp et al. (2022) indicate MESHS resolves the 28 June 2021 event as having a very high rank in history, and they describe MESHS return periods above 70 to 100 years. This suggests radar-based hail does resolve high hazard severity, therefore, the low estimated loss for this severe event must be due to the model's impact functions. This counter-evidence is just one event, however, inspection of the top ten observed events (focusing on the top right corners of plots in Figure 8) shows the model has a low bias for these most severe events, hence the model's impact functions could be responsible for those modelled loss errors which are most material to companies (responding to large hail events). If radar-based hail does resolve high hazard severity, then I can think of two reasons why impact functions fail to convert high hazard to severe loss, as now discussed.

First, the horizontal drift of hail from radar levels to the ground means the calibration of damage *per building* cannot be sharply resolved by radar data. For example, high MESHS will often be collocated with buildings experiencing smaller hail at the ground, and vice-versa, leading to a much weaker relation between MESHS and damage ratio than actually exists. This possibility may be worth discussion in the text?

A potential second source of weak relation from hazard to damage is the fact that the damage ratio varies with building attributes. The authors explore some possibilities in lines 462 to 470 and find some weak evidence. Have the authors considered splitting buildings by rural and urban settings, and building separate impact functions? Much higher damage ratios in rural

regions are a significant feature in the U.S., though I don't know if this applies to Switzerland too.

To be clear, I ask that these possibilities be considered for discussion in the manuscript, while any further model development is at the authors' discretion.

We thank the reviewer for these comments. We structured our response in three parts.

Part 1: MESHS as main source of uncertainty
We agree with the reviewer that on a storm-footprint level MESHS does resolve extreme events well, at least in relative terms (i.e. events with extreme damages mostly also have areas with high MESHS values, though not consistently co-located). Thus, the calibrated impact functions provide reasonable results on an aggregated scale (i.e. the model does resolve extreme events on aggregated scales). However, within storm footprints high MESHS values are not consistently correlated with damages.

For example, on 28 June 2021, high MESHS values (>70mm; Fig. 10) with expected return periods >100 years according to Kopp et al. (2022, their Fig. 5) were observed in a patch over southern Zürich and in a second patch over northern Zürich. The high MESHS values in the southern patch are co-located with high building damages (Fig. 10). Contrastingly, the high MESHS in the northern patch are co-located with no or only small building damages (Fig. 10) despite high building density (Fig. 2). Crowdsourced data show that while there was some hail in the northern patch, it is largely labelled as 'Coffee bean' (<5mm)' or 'One franc coin' (23mm) size (Appendix B), which is a sharp contrast to the estimated >100-year return period based on MESHS in Kopp et al. (2022). Thus, based on this exemplary case, we may speak of "high MESHS with damages" (southern patch) and "high MESHS without damages" (northern patch; see also discussion of datapoints generating high FAR regarding Fig. 4 and Fig. 5).

It is important to note that both kinds of events, MESHS with damage and MESHS without damage, influence the calibrated impact function. In case of the empirically calibrated impact function, the damage estimate assigned to high MESHS represents the average of "high MESHS with damages" and "high MESHS without damages". While the reviewer's statement that "the low estimated loss for this severe event must be due to the model's impact functions" is true, it ignores that the impact function is a direct result from the spatial match between MESHS and damages. If we had less "high MESHS without damages", our impact function could better represent events with extreme damages. Based on these considerations, we arrived at our statement that MESHS is currently the factor limiting model skills the most. As MESHS is related to the storm intensity and represents the *expected* hail size, rather than directly measuring the size of individual hail stones (e.g. Schröer et al, 2023), this result is not completely unexpected.

Part 2: Taking into account horizontal drift of falling hailstones

While on 28 June 2021 horizontal hail drift cannot explain the spatial mismatch for the northern patch of "MESHS without damages" (which is >10km displaced from areas with intense hail damages or crowdsourced reports of extreme hail), we acknowledge that hail drift is a factor which reduces model skill and may be partly responsible for the low-intensity-large-area bias of the model. Thus, an explicit hail drift which maximizes the correlation between MESHS and relative damages was newly implemented after considering the reviewer's comments and found not to increase verification skill scores strongly. Thus, the partial mismatch in areas of extreme MESHS and extreme damages can only to a small fraction be explained by wind-driven hail drift. **See Comment C28 for more details and new Fig. 4**. Note that in addition to the verification skill scores, model skill scores using the shifted MESHS were obtained (not shown) and similarly showed only minimal improvement *(1-2% better $POD_M$, $FAR_M$, $FH_M$)*.

Part 3: Vulnerability depending on building attributes

Indeed, attributes such as age, year of construction, used materials, roof area, or type of blinds have a large impact on vulnerability of an individual building. With accurate data of both the hazard (hail) and exposure (buildings), accounting for these factors would also improve model estimates on an aggregated (e.g. storm footprint) scale. However, with the available hazard data (i.e. MESHS), which cannot be directly interpreted as actual local hail size (cf. Part 1 of this answer, Section 5.1 in the paper, Schröer et al., 2023), an accurate quantification of building attribute-dependent vulnerability remains challenging.

In our data, the year of construction has a clear impact on vulnerability (almost factor 2) and so does building volume (Fig 3, Comment C8), which is strongly correlated with rural regions which have fewer large multistory buildings. Both factors were considered: The volume with a scaling in the paper the year of construction not shown, but discussed in Line 465ff.
However, neither improved the skill scores significantly because with the current hazard data, only damage estimates within 1 OOM are feasible. Thus, even a factor 2 is comparably small to the current precision of the MESHS-based hail damage model. Furthermore, a factor 2 on a per-building-level translates to a much smaller difference on an aggregated scale such as one storm footprint which, for which the model is suitable. Lastly, impact functions depending on building attributes cannot be used with gridded exposure data, which would limit the model applicability strongly.

In addition to the existing discussions of the influence of variable building vulnerability below:
- L463-466: An impact function represents damages on aggregated spatial scale.
- L466-468: Older buildings have lower vulnerability.
- 157-163, Table 2, L469-470: Value-dependent "scaled" building exposure, which aims to correct the damage overestimation for large/expensive buildings, does not improve model skills notably.

We add the following sentences to the manuscript:

L207/L452: Explaining the explicit hail drift and the implications of the limited improvement in verification skill scores from unidirectional hail drift. **See comment C28 for implemented (major) changes and updated Fig. 4.**

L232: *The consistently observed damages within a hail streak and the minimal improvement in FAR with explicit hail drift indicate that the false alarms are largely related to limitations in radar-based hail proxies rather than variable building vulnerability or wind-driven hail drift.* **See comment C14 for details**

L471: *While the imprecise nature of available radar metrics with the high FAR (Sect. 3) makes a direct quantification of the influence building attributes challenging, the* lack of model skill improvement when considering more details of the exposure data  *suggests* that currently the overall large uncertainty in the hail damage model mainly stems from the available  radar metrics. Since none of the considered *single-polarization* radar variables accurately and consistently distinguishes large hail from smaller (not damaging) hail, damage estimates have high uncertainty.

**Minor comments/corrections**

**C26:** line 16: could the damages be expressed in EUR, either as a replacement or as an addition to
CHF? A significant fraction of readers may have to break away from the manuscript after the first
sentence to check the value of a CHF.

We thank the reviewer for this comment and address it as described in Comment C4 of reviewer 1.

**C27:** In Section 2.2.1, could an annual timeseries of number of claims per year, and mean claim size
(both normalised) be included? It may help to understand why model validation in Table 2 is slightly
poorer in the first ten years – perhaps higher FAR is due to poorer data collection 10-20 years ago?

We thank the reviewer for this comment. We do not find any gaps, jumps, or trends in the timeseries of normalized number and mean size of claims (see figure below) that suggest lower data collection quality during the early years of the considered period. Furthermore, we would like to highlight that lower model skill during 2002-2012 is expected because of the lower quality of the hazard data during this period (see L115ff and Trefalt et al., 2023). We have no reason to believe that the damage data quality further deteriorates the model skill. Therefore, we do not include the figure below or adapt the text of the manuscript. Also see comment C31 regarding the radar data accuracy before/after 2012.

[Figure]

Fig R3: Yearly timeseries of the number of reported damage claims (blue) and the average (orange) and median (orange dotted) damage sum per claim.

**C28:** Figure 4, and text: results using the 4 km buffer could be viewed as more appropriate, whereas
those for 1 km cells give the reader an indication of how much hail drift can affect the relation
between hail at radar levels and damage on the ground. From this viewpoint, the solid and dashed
lines could be swapped in the top row of Figure 4, and the text in Section 3 focuses more on the 4
km buffer data? Irrespective of whether the 4 km buffer is viewed as the standard choice, could the
HSS for 4 km buffer data be included in Figure 4?

We thank the reviewer for this comment. As the 4km buffer is applied differently for the $POD_{4km}$ and $FAR_{4km}$ (Line 209-211), the HSS cannot be directly calculated from the two values.
For POD, any gridcell with damage reports and with 'expected hail' (i.e. hazard value >threshold) within 4km is considered a hit, which reduces misses. For FAR, any gridcell

with 'expected hail' is considered a hit if there is a reported damage within 4km, which reduces false alarms. Both values correct for hail drift, but overcompensate. The false alarms in the FAR$_{4km}$ are cells with *certainly* no damaging hail (no report within 4km) despite large hazard values. However, as hail drift does not occur in all directions equally, more gridcells will be considered hits than could realistically be reached by hail drift. Thus, the FAR_4km is interpreted as a 'minimum possible FAR' considering a 4km hail drift in all directions.

Upon reflection on the reviewers' comments, we decided that an explicit consideration of unidirectional hail drift will be more intuitive for the reader. By shifting the hazard footprint up to 4 km each day to maximize the correlation between hazard data and the observed proportion of damaged buildings, we created a shifted hazard set. This dataset accounts for hail drift without overcompensating. As the highest damages are mostly concentrated one hail streak per day, we only consider one hail drift vector per day. This method does not fully account for the variable hail drift within a storm, but captures cases where a wind-driven displacement in one direction is observed.

The resulting skill scores (see Figure below) show a much weaker improvement than the 4km buffer, which confirms that the good skill scores with a 4km buffer are only to a small extent explained by unidirectional hail drift (but rather by the smoother spatial structure of MESHS in comparison to hail damages). Note that a hail drift of up to 8km and an alternative method of maximizing the HSS instead of the correlation were tested without qualitatively changing the result.

[Figure]

Figure 4.  *(a,b,c)* MESHS, maximum reflectivity, and E$_{kin}$ verification statistics probability of detection (POD), false alarm ratio (FAR), and Heidke-skill-score (HSS) calculated per gridcell for all 1km cells with >10 buildings. *Dashed lines indicate the same skill scores with post-processed hazard data, which explicitly captures unidirectional hail drift by maximizing the correlation to observed relative damages (refer to text for details).* Grey bars show the number of gridcells with given hazard intensity over the whole time period with a logarithmic scale.  *(d,e,f)* Visualization of contingency table variables per gridcell for a single event (28 June 2021) and selected hazard thresholds corresponding to the maximum HSS. Grey areas do not contain sufficient exposure data (<10 buildings) and white areas mark cells with missing hazard data.

In the paper we adjust the following sections focusing on unidirectional hail drift instead of a 4km buffer:

L208: ~~Thus, both POD and FAR are also shown with a 4 km buffer which accounts for horizontal drift of hail (Barras et al., 2019). For the POD4km, a gridcell with a damage report is considered a hit if the hazard threshold is reached anywhere within 4 km. For the FAR4km, a gridcell which exceeds the hazard threshold (i.e. hail damage expected) is only considered a false alarm if there is no damage report within 4 km.~~ *Because a part of the increased skill at lower resolution could be explained by hail drift, we additionally calculate skill scores with an explicit hail drift considered. Hazard data for each day are artificially shifted to maximize the correlation to the percentage of affected buildings in all gridcells with 10 or more buildings. As the highest damages are mostly concentrated in one hail streak, we only consider one hail drift vector per day with a maximum length of 4km accounting for the 2–4 km wind drift of hailstones (Hohl et al., 2002b; Barras et al., 2019). Skill scores obtained from the shifted hazard data are shown for comparison in Fig. 4 and consistently show a marginal improvement over the original skill scores. Given the minimal improvement, but inability to consider damage data-dependent hail drift in real-time applications, we focus on the non-shifted hazard data for model development.*
At the minimum MESHS value of 20mm, the POD reaches 60% for an exact spatial match , indicating that 60% of all gridcells with a hail damage report lie within a MESHS footprint. When analyzing individual claims (not shown), detection probabilities of 88%  are reached.

L215: Thus, even at extreme MESHS values only one in two gridcells with >10 buildings contains a damage claim and in 20% of cases there is not a single damage claim within 4 km *(not shown).*

L225: Thus, the POD decreases rapidly while the FAR decreases linearly with increasing $E_{kin}$ reaching 60% at 1000Jm−2 , comparable to the FAR of MESHS at 80mm.

L232: **See comment C14**

L452: **Horizontal hail drift.** Some studies explicitly consider horizontal drift and shift radar footprints to obtain an optimal overlap to insurance claims, which improves correlations (Hohl et al., 2002a; Schuster et al., 2006).  *Here, we consider this hail drift in the insurance claims pre-processing, and by analyzing hail occurrence skill scores with a correlation-maximizing unidirectional shift of hazard variables (Sect. 3).* For the model calibration and evaluation,  no spatial shift is considered for two main reasons. *First, the explicit hail drift only marginally improves skill scores in Fig. 4 which* suggests that MESHS generally spatially overpredicts hail streaks on the ground, rather than suffering from wind related directed shifts. *Secondly, most insurance claims (88%) are within a MESHS footprint without considering a spatial shift.*
 *Furthermore,* an optimal shift (as in Hohl et al., 2002a; Schuster et al., 2006) could not be applied in a real-time application, before damage claim data is available.

**C29:** Figures 6 and 7, and text: could the CDR be included in these two figures? The CDR is the damage ratio, conditional on damage occurring, and often referred to as mean severity. CDR is expected to rise with increasing hazard severity but it is not clear this is the case from figures 6a and b, or 7a and b.

We thank the reviewer for this comment. As part of the analysis, we indeed calculated the CDR, which is labelled 'Mean Damage Degree (MDD)' within CLIMADA. Between reporting the total loss ratio or 'Mean Damage Ratio (MDR) and the CDR/MDD, both of which have previously been used to quantify hail damages, we chose to report the MDR for the following main reasons:

Firstly, the CDR/MDD is prone to an interpretation on an individual building-level as it reports the relative damage, given that a damage claim was made. As our model is not suitable to estimate building-level damages, but rather aggregated damages over many buildings, we found the MDR (which additionally considers the ratio of affected buildings) a better choice. Secondly, in many cases a model user may only have access to gridded exposure data of total building values, in which case they can directly use the MDR impact function.

Nevertheless, we agree with the reviewer that mentioning the mean severity, which increases strongest above 50mm MESHS, would be interesting for some readers and added the following sentence:

L304: For the MDR, the empirical fit exhibits a steeper increase between 50 and 60mm, which cannot be fully captured by the chosen sigmoidal fit. *While the increase in MDR up to 50mm is largely driven by increasing PAA, the sharp increase above 50mm MESHS occurs due to a strong increase in mean severity partly caused by extreme (roof-penetrating) damages. Mean severity quantifies the damage ratio, given that a damage claim was made. Thus, the mean severity multiplied with the PAA corresponds to the MDR, assuming that the average value of damaged buildings corresponds to the average value of all buildings.*

**C30:** x-axis legend on figure 8d: typo in 'normalized'
We thank the reviewer for this comment and fixed the typo.

**C31:** line 351: any evidence that the latest radar has fewer small patches? Or is there more complete
data collection in the present-day? (Related to comment 2 above.)
We thank the reviewer for this question. While from communication with the data providing building insurances, we have no reason to expect less complete data collection before 2012 (also see C27), reduced radar data quality in this time period is documented in Trefalt et al. (2023). While the improved automated pre-processing with the new generation (dual-pol) radars is mentioned in Sect. 2.1, an explicit quantification of small patches throughout the considered time period is not shown in the paper. Thus, we rephrase the sentence as follows:

L351: Many of these false alarms are days with multiple few-km-wide patches of positive MESHS values  *but few or no damage reports. After 2012, the fraction of false alarms reduces, which is likely explained by improved automatic pre-processing with dual-polarization radars (Trefalt et al., 2023; Sect 2.1).*

Trefalt et al. (2023) https://doi.org/10.18751/PMCH/TR/284.HailClimateSwitzerland/1.0

**C32:** Figure 10: the reader may get more benefit from maps of damage ratio values in the 2nd and 3rd columns, with total loss remaining in the title? The damage ratio reflects MESHS values and might
be more interesting than loss values which mainly reflect exposure density.
We thank the reviewer for this comment. After careful consideration, we decided to keep the absolute hail damages rather than the damage ratio in Figure 10. Our decision is based on the following considerations: Firstly, showing absolute damages allows the reader (and potential model user) to see how a model output for one hail event looks like. In contrast, the modelled damage ratio would appear as a rescaled MESHS footprint, according to the calibrated impact function (Fig. 6). Secondly, the dependence of hail damage estimates on a specific (possibly

user-provided) exposure layer is an important aspect of the model, which we would like to highlight here.

**C33:** Lines 471-474: results in Table 2 show MESHS clearly distinguishes damaging events (> 100
claims) from non-damaging events. The problem for most users will be that the loss model does not
resolve the most severe events in history, and as mentioned above in the second Main Comment,
MESHS may resolve high hazard severity but they are not translating to high loss.

We thank the reviewer for this comment. While we agree with the reviewer that the model tends to underestimate the most severe events, we argue that the model still 'resolves' these events. Model skill in terms of $FH_M$ (fraction of events with the correct order of magnitude) as well as in terms of the average model-to-observed ratio (see comment C24) is not worse (but actually better) for the most extreme events compared to medium-impact events (Fig. 8). However, the model error is more systematic as it largely predicts lower damages for the few most severe events. Thus, given a model estimate of both a large impact (e.g. 20'000 damage claims) or a medium impact (500 damage claims) we can expect a likely actual damage within 1 OOM of the model estimate. While for the large impact the actual damage is more likely to be lower rather than higher, the expected ratio of modelled and observed damages is similar for the large and small predicted impact (i.e. the 'modelled-to-observed factor' in C 24).

In contrast to the event-wise losses, the extreme local damages within one hail swath are not resolved in the model (also see C25 and Sect 5.1 Line 392ff). While high MESHS does often occur on days with extreme damages, there are also areas with high MESHS but no observed damages despite exposed buildings (FAR in Fig. 4/5, and exemplary in Fig. 10, particularly on 28 June 2021). Thus, one can argue that MESHS does resolve high hazard severity, but it does so too often (high FAR) and therefore the impact function cannot attain as high relative damages as reported. If the model was calibrated without considering the false alarms, local damages would be more realistic, but total damage estimates would be systematically overpredicted. This argument is explained in more detail in comment C25.